# FDA-Approved Small Molecule Compounds as Drugs for Solid Cancers from Early 2011 to the End of 2021

**DOI:** 10.3390/molecules27072259

**Published:** 2022-03-31

**Authors:** Aleksandra Sochacka-Ćwikła, Marcin Mączyński, Andrzej Regiec

**Affiliations:** Department of Organic Chemistry and Drug Technology, Faculty of Pharmacy, Wroclaw Medical University, 211A Borowska Street, 50-556 Wroclaw, Poland; marcin.maczynski@umw.edu.pl (M.M.); andrzej.regiec@umw.edu.pl (A.R.)

**Keywords:** anticancer drugs, small molecule agents, solid cancers, approval characteristic, FDA, EMA

## Abstract

Solid cancers are the most common types of cancers diagnosed globally and comprise a large number of deaths each year. The main challenge currently in drug development for tumors raised from solid organs is to find more selective compounds, which exploit specific molecular targets. In this work, the small molecule drugs registered by the Food and Drug Administration (FDA) for solid cancers treatment between 2011 and 2022 were identified and analyzed by investigating a type of therapy they are used for, as well as their structures and mechanisms of action. On average, 4 new small molecule agents were introduced each year, with a few exceptions, for a total of 62 new drug approvals. A total of 50 of all FDA-approved drugs have also been authorized for use in the European Union by the European Medicines Agency (EMA). Our analysis indicates that many more anticancer molecules show a selective mode of action, i.e., 49 targeted agents, 5 hormone therapies and 3 radiopharmaceuticals, compared to less specific cytostatic action, i.e., 5 chemotherapeutic agents. It should be emphasized that new medications are indicated for use mainly for monotherapy and less for a combination or adjuvant therapies. The comprehensive data presented in this review can serve for further design and development of more specific targeted agents in clinical usage for solid tumors.

## 1. Introduction

Cancers are one of the major causes of human death throughout the world each year. In 2020, the five solid cancer leaders of mortality were lung cancer (18.0% of the total cancer deaths), colorectal cancer (9.4% of the total cancer deaths), liver (8.3% of the total cancer deaths), stomach cancer (7.7% of the total cancer deaths) and female breast cancer (6.9% of the total cancer deaths) [1]. Solid cancers, which are classified as sarcomas or carcinomas, are diseases characterized by an abnormal mass of certain tissues. Sarcomas predominantly develop from the embryonic mesoderm and can occur anywhere in the body, for example in bones, nerves and soft tissues [2]. In contrast, carcinomas arise from the epithelium, found in the skin or the lining of the body’s internal organs, such as the stomach, breast or lung [3]. The cancerous cells have the ability of unregulated proliferation and can invade surrounding tissues or spread to more distant ones via the bloodstream or lymphatic system. Tumor growth and metastasis spread depend on angiogenesis, which is a process of formation of new blood vessels triggered by signaling molecules in rapidly dividing cancerous cells [4]. Small molecule drugs can lead to solid cancer regression both by direct inhibition of cancerous cells proliferation and angiogenesis suppression. They may be used alone or in combination with other therapeutic agents or treatment options, such as surgery, radiation therapy and immunotherapy. The use of a combinational regimen is rational clinically to enhance the treatment efficacy and to avoid the progression of therapy resistance. The result of combination therapy is an achievement of a long-lasting, high response rate and an improvement in the overall survival of the patients compared to using single-agent treatment [5]. 

This article is an overview of small molecule compounds used in the treatment of solid cancers, which obtained FDA approval from early 2011 to the end of 2021. Both the drugs containing a new molecular entity and known active agents, but in the new formulation, are outlined in the present review. The registered small molecules are presented and discussed, focusing on the initial approval date, chemical structure, molecular target, route of administration, indication and most frequent adverse effects. An effort was made to analyze the approved drugs in terms of the type of therapy they are used for (i.e., monotherapy, combination therapy or adjuvant treatment), their mechanism of action (targeted, cytostatic, radioactive or hormone therapy) and the pharmacophores contained in their structures (aromatic rings, heteroatoms and functional groups). Notably, drugs, which obtained supplemental indications in the period from 2011 to 2022 but were originally approved before 2011. Additionally, drugs, which are used to treat the side effects of cancer therapy or to identify cancer diseases are not included in this work.

## 2. Various Protein Kinase Inhibitors as Anticancer Agents

Protein kinases (PTKs) are enzymes that regulate the biological activity of proteins by phosphorylation of certain amino acid residues. This reaction causes a conformational change from an inactive to an active form of the protein, which is one of the most important regulatory mechanisms of the cell cycle and transduction of external signals. Dysregulation of protein kinases activity is implicated in the processes of carcinogenesis and the progression of various solid cancers. Therefore, protein kinases are prime targets for the development of selective anticancer drugs [6].

### 2.1. Tyrosine Kinase (TK) Inhibitors

Tyrosine kinases (RTKs) are enzymes that selectively phosphorylate the hydroxyl groups of a tyrosine residue in different proteins with adenosine triphosphate (ATP) as the source of phosphate. They have a share in the regulation of the most fundamental cellular processes, such as growth, differentiation, proliferation, survival, migration and metabolism of cells or programed cell death in response to extracellular and intracellular stimuli [7]. There are two types of tyrosine kinases, namely receptor tyrosine kinases (RTKs) and nonreceptor tyrosine kinases (NRTKs) [8]. A lot of RTKs and NRTKs are associated with cancers, thus a significant number of tyrosine kinase inhibitors (TKIs) are currently in clinical development. Since 2011, the FDA approved eleven new anticancer drugs that are inhibitors of anaplastic lymphoma kinase (ALK), epidermal growth factor receptor (EGFR or HER1), human epidermal growth factor receptor 2 (HER2), human epidermal growth factor receptor 4 (HER4), fibroblast growth factor receptors (FGFRs), vascular endothelial growth factor receptors (VEGFRs), mesenchymal-epithelial transition factor (MET) or receptor tyrosine kinase rearranged during transfection (RET) (Table 1). These drugs show anticancer activity by blocking multiple molecular signal transduction pathways (Figure 1).

The oncogenic driver mutations identified in non-small-cell lung cancer (NSCLC) include ALK gene rearrangements, ROS1 gene rearrangements, EGFR mutations, MET mutations and RET rearrangements [16]. In NSCLC harboring ALK gene rearrangements are observed ALK fusion proteins with potent transforming activity as oncogenic drivers of tumor growth [17]. **Ceritinib** is the second-generation AKL inhibitor that blocks autophosphorylation of ALK and ALK-mediated phosphorylation of signal transducer and activator of transcription 3 (STAT3), which is a downstream signaling protein [18,19]. Hence, this drug inhibits the cell cycle in the G1 phase and the proliferation of ALK-dependent cancer cells. Among the existing therapies targeting EGFR-mutated NSCLC, there have been two FDA-approved medicaments during the last eleven years, i.e., **osimertinib** and **mobocertinib**. **Osimertinib** is a third-generation, irreversible TK inhibitor of both EGFR TKI-sensitizing mutations and a secondary EGFR mutation in exon 20, namely T790M [20]. **Mobocertinib**, on the other hand, is a first-in-class irreversible EGFR TK inhibitor, which was specifically developed to selectively inhibit oncogenic variants containing EGFR exon 20 insertion (EGFRex20ins) mutations. Both drugs form a covalent bond with cysteine 797 in EGFR with high-affinity binding resulting in sustained EGFR activity inhibition [21,22]. The difference in the structure of these drugs is the presence of an isopropyl ester group on the pyrimidine ring of **mobocertinib,** leading to increased selectivity for the EGFRex20ins mutant compared with **osimertinib** [22]. In NSCLC, MET and its mutant variants produced by gene mutation, amplification and overexpression are attractive targets for a blockade. For example, MET and variant with exon 14 skipping mutation are targets for **capmatinib** and **tepotinib** activity. The drugs act by inhibition of MET phosphorylation and the activation of key downstream effectors in MET-dependent cancer cell lines [23,24]. The cancers harboring RET alterations, particularly NSCLC, can be treated with **pralsetinib**. It selectively inhibits RET autophosphorylation and proliferation of RET-mutant cancer cells [22].

Overexpression of HER2 occurs approximately in 15 to 20% of breast cancers. **Neratinib** and **tucatinib** are inhibitors of the human epidermal growth factor receptors (HERs) that are used for the treatment of HER2-positive breast cancer (HER2 + BC). **Neratinib** irreversibly inhibits EGFR, HER2 and HER4 kinases, while **tucatinib** reversibly and highly selectively blocks HER2. The drugs have shown to be effective in monotherapy or in combination chemotherapy with **capecitabine** [25,26]. Patients with HER2 + BC who have disease progression after prior therapy with multiple HER2-targeted drugs may benefit from these TKIs used with or without **trastuzumab** [27,28]. The mechanism of action of both drugs includes binding to the ATP pocket of the HER2, which results in decreased receptor autophosphorylation and inhibition of downstream mitogen-activated protein kinase (MAPK) and phosphatidylinositol triphosphate kinase (PI3K) signaling. This leads to cell cycle arrest at the G1-S phase, thereby reducing cell proliferation [29,30].

FGFR2 fusion or rearrangements are present in 10–16% of intrahepatic cholangiocarcinomas. Treatment options, which improve clinical outcomes of patients with cholangiocarcinoma (CCA) harboring FGFR2 gene fusions, have been extended to the first two targeted therapies, i.e., **pemigatinib** and **infigratinib** [31,32]. The FDA approval of these TKIs includes the indication for adults with previously treated, unresectable, locally advanced or metastatic CCA. Their mechanism of action is a selective, ATP-competitive inhibition of fibroblast growth factor receptors (FGFRs). Both drugs potently inhibit FGFR1, FGFR2 and FGFR3 kinases and also demonstrate weaker activity against FGFR4 [33,34].

Renal cell carcinoma (RCC) is the most common type of kidney cancer. From a pathologist’s point of view, RCC tends to be a highly vascular tumor. The prominent vascularization is due to the increased production of proangiogenic growth factors, such as vascular endothelial growth factor receptors (VEGFRs) [35]. **Tivozanib** is a quinoline-urea derivative that inhibits VEGFRs in an ATP-competitive manner. In particular, the drug shows inhibitory activity against VEGFR-1, VEGFR-2 and VEGFR-3 at picomolar concentrations. The analysis of the mechanism of action indicates that **tivozanib** produced a significant inhibition of the ligand-induced phosphorylation of VEGFRs causing direct anticancer activity as well as suppression of angiogenesis and vascular permeability [36]. In clinical trials, this agent used as third-line or fourth-line therapy in patients with RCC improved progression-free survival and was better tolerated than sorafenib [37]. The promising results of **tivozanib** led to its approval by the FDA for the treatment of adult patients with relapsed or refractory advanced RCC following two or more prior systemic therapies [38].

**Table 1 molecules-27-02259-t001:** Features of the tyrosine kinase inhibitors approved as drugs by the Food and Drug Administration (FDA) from 2011 to 2022. The order of drugs is tabulated in order of most recent to oldest registration date. A generic name of a drug is an international nonproprietary name (INN).

No	Generic Name of Drug	Brand Name and Company	First FDA/EMA Approval Date	Structure	Molecular Target	Route of Administration	Indication	Adverse Effects	Ref.
1	Mobocertinib	EXKIVITY Takeda Pharmaceuticals America, Inc., Deerfield, IL, USA	FDA: 15 September 2021 EMA: Not approved	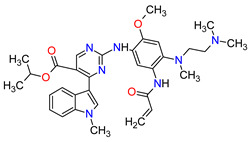	EGFR ^1^	Oral	Non-Small Cell Lung Cancer	Diarrhea, rash, stomatitis, vomiting, decreased appetite, nausea, paronychia, musculoskeletal pain, dry skin, fatigue, decreased hemoglobin, decreased lymphocytes, increased creatinine, amylase, and lipase, decreased potassium, and magnesium	[39]
2	Infigratinib	TRUSELTIQ BridgeBio Pharma, Inc., Palo Alto, CA, USA	FDA: 28 May 2021 EMA: 21 August 2020	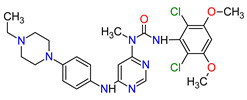	FGFRs ^2^	Oral	Cholangiocarcinoma	Nail toxicity, stomatitis, dry eye, fatigue, increased creatinine, phosphate, alkaline phosphate, and alanine aminotransferase, decreased phosphate, and hemoglobin	[40,41]
3	Tivozanib	FOTIVDA AVEO Oncology, Boston, MA, USA; Eusa Pharma (Netherlands) B.V., Schiphol-Rijk	FDA: 10 March 2021 EMA: 24 August 2017	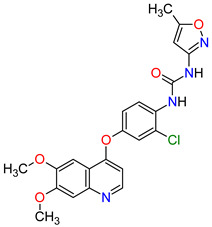	VEGFRs ^3^	Oral	Renal Cell Carcinoma	Fatigue, hypertension, diarrhea, decreased appetite, nausea, dysphonia, hypothyroidism, cough, stomatitis, sodium decreased, lipase increased, and phosphate decreased	[38,42,43]
4	Tepotinib	TEPMETKO EMD Serono, Inc., Darmstadt, Germany.	FDA: 3 February 2021 EMA: Not approved	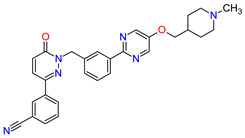	MET ^4^	Oral	Non-Small Cell Lung Cancer	Peripheral edema, diarrhea, fatigue, nausea, decreased appetite, increased blood creatinine levels, hypoalbuminemia, increased amylase levels	[44]
5	Pralsetinib	GAVRETO Genentech, Inc., South San Francisco, CA, USA	FDA: 4 September 2020 EMA: 18 November 2021	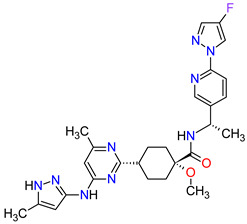	RET ^5^	Oral	Non-Small Cell Lung Cancer	Fatigue, constipation, musculoskeletal pain, hypertension	[45,46]
6	Capmatinib	TABRECTA Novartis Pharmaceuticals Corporation, Basel, Switzerland	FDA: 6 May 2020 EMA: Not approved	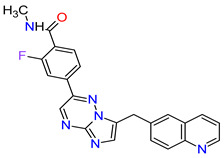	MET ^4^	Oral	Non-Small Cell Lung Cancer	Peripheral edema, nausea, fatigue, vomiting, dyspnea, decreased appetite	[47]
7	Pemigatinib	PEMAZYRE Incyte Corporation, Wilmington, DE, USA	FDA: 17 April 2020 EMA: March 26, 2021	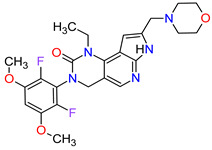	FGFRs ^2^	Oral	Cholangiocarcinoma	Hyperphosphatasemia, alopecia, diarrhea, fatigue, dyspepsia	[48,49]
8	Tucatinib	TUKYSA Seattle Genetics, Inc., Bothell, WA, USA	FDA: 17 April 2020 EMA: 11 February 2021	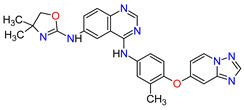	HER2 ^6^	Oral	Breast Cancer	Diarrhea, palmar–plantar erythrodysesthesia syndrome, decreased hemoglobin or phosphate, nausea	[50,51]
9	Neratinib	NERLYNX Puma Biotechnology, Inc., Los Angeles, CA, USA	FDA: 17 July 2017 EMA: 31 August 2018	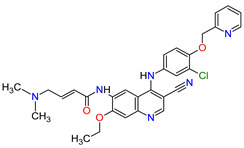	EGFR ^1^, HER2 ^6^, HER4 ^7^	Oral	Breast Cancer	Diarrhea	[52,53]
10	Osimertinib	TAGRISSO AstraZeneca, Cambridge, UK	FDA: 13 November 2015 EMA: 24 April 2017	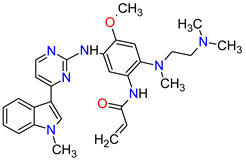	EGFR ^1^	Oral	Non-Small Cell Lung Cancer	Diarrhea, rash, dry skin, nail toxicity	[54,55]
11	Ceritinib	ZYKADIA Novartis Pharmaceuticals Corporation, Basel, Switzerland	FDA: 29 April 2014 EMA: 6 May 2015	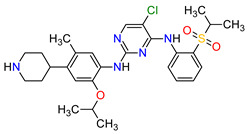	ALK ^8^	Oral	Non-Small Cell Lung Cancer	Diarrhea, nausea, vomiting, abdominal pain	[56,57]

^1^ **EGFR**: epidermal growth factor receptor. ^2^ **FGFRs**: fibroblast growth factor receptors. ^3^ **VEGFRs**: vascular endothelial growth factor receptors. ^4^ **MET**: mesenchymal-epithelial transition factor. ^5^ **RET**: tyrosine kinase rearranged during transfection receptor. ^6^ **HER2**: human epidermal growth factor receptor 2. ^7^ **HER4**: human epidermal growth factor receptor 4. ^8^ **ALK**: anaplastic lymphoma kinase.

### 2.2. Cyclin-Dependent Kinase (CDK) Inhibitors

Cyclin-dependent kinases (CDKs) are a family of serine-threonine kinases. They regulate the cell cycle and other important cellular functions, including gene transcription, metabolism and neuronal function. The human genome codifies 20 CDKs (1–20) and 13 groups of cyclin, which are proteins that control the activities of the CDKs through their oscillating level during the cell cycle [58]. The CDKs have been grouped into cell cycle-related subfamilies (CDK1, 4 and 5) and transcriptional subfamilies (CDK7, 8, 9, 11 and 20). Dysregulating the CDKs and cyclins level leads to abnormal cell proliferation and tumor growth. Owing to the role of CDKs in cancer cells, their inhibition is an important target for novel anticancer drugs. The suppression of CDK4 and CDK6 activity is now being investigated to treat various solid tumors, including lung, prostate and ovarian cancers. The CDK4/6 inhibitors, i.e., **palbociclib**, **ribociclib** and **abemaciclib**, demonstrated promising clinical activity in the treatment of advanced breast cancer, thereby being recently FDA approved (Table 2) [59,60]. The approval of **abemaciclib** (as VERZENIO) includes using it for monotherapy or in combination with **fulvestrant,** which is an estrogen receptor antagonist. **Palbociclib** (as IBRANCE) was registered for combination therapy with **fulvestrant** or an aromatase inhibitor (**letrozole**). **Ribociclib** (as KISQALI) was approved only in combination with an aromatase inhibitor (**letrozole**) for initial endocrine-based therapy. All of these drugs are selective inhibitors of cyclin-dependent kinase 4 (CDK4) and 6 (CDK6). They inhibit Rb protein phosphorylation in the early G1 phase, thereby blocking cell-cycle transition from G1 to S phase and reducing cancer cell growth (Figure 2) [61,62,63,64]. In addition, **abemaciclib** is able to penetrate the blood–brain barrier (BBB), thereby being promising to achieve regression of primary and metastatic tumors involving the central nervous system (CNS). The ongoing clinical trials allow evaluating **abemaciclib** for therapy in patients with brain metastases and leptomeningeal metastases (LM) secondary to HR-positive breast cancer [65,66].

### 2.3. Multi-Kinase Inhibitors

In solid cancers, the frequently aberrant activity of various components of signaling pathways occurs by the hyperactivation of several different kinases (Figure 3). Multi-kinase inhibitor is one agent that targets a set of structurally related kinases leading to simultaneous blocking of their activity [73]. The use of one multi-kinase inhibitor is preferred to two single agents, since drug–drug interactions can trigger changing metabolism and activities against particular kinases. Multi kinase drugs become the second choice when their pharmacokinetic properties are worse. Besides, multi-kinase inhibitors are less specific and can consequently result in more side effects. The disadvantage during treatment with multi-kinase inhibitors is acquired resistance [74]. The approval characteristics of FDA-registered multi-kinase inhibitors are presented in Table 3.

Patients with NSCLC receiving the first-generation TKIs, e.g., **crizotinib**, **geftinib** and **erlotinib**, experienced issues related to acquired resistance. This resistance can develop by various mechanisms, such as **crizotinib**-resistant mutations in the anaplastic lymphoma kinase (ALK) domain. In addition, patients’ treatment with **crizotinib** often develops CNS metastases, likely due to the poor CNS penetration of **crizotinib**. However, **crizotinib** exhibits higher clinical response rates than standard chemotherapy and is recommended both for first-line therapy in NSCLC, as well as next-line therapy in patients who have not been treated with **crizotinib** previously. The next-generation multi-kinase inhibitors are designed to overcome TKI-resistant mutations. **Alectinib**, **brigatinib** and **entrectinib**, which are the second-generation ALK inhibitors, possess activity against treatment-resistant ALK mutants, whereas **lorlatinib**, which belongs to the third-generation drug, is highly selective proto-oncogene tyrosine-protein kinase ROS (ROS1) and ALK inhibitor and has the ability of robust brain penetration [75]. The second-generation EGFR TKIs, namely **dacomitinib** and **afatinib**, are characterized by their broader activity against HER family members and irreversibility, covalently binding to their targets of the kinases domain. They have the potential for anticancer activity against receptors with acquired mutations that are resistant to first-generation inhibitors. For example, **dacomitinib** specifically inhibits EGFR with exon 19 deletion or exon 21 L858R substitution mutations but also inhibits HER2, HER4 and transphosphorylation of HER3 [76]. 

Fibroblast growth factor receptor (FGFR) mutations are frequently observed in a variety of malignancies, e.g., FGFR2/3 alternations are common in urothelial carcinoma. **Erdafitinib**, a pan-FGFR inhibitor, is a promising therapy for cancers harboring these mutations. **Erdafitinib** obtained its first global approval in 2019 for the treatment of adult patients with locally advanced or metastatic urothelial carcinoma with FGFR alterations. The response to treatment was fast and independent of the number of previous therapies, the presence of visceral metastasis or tumor location [77]. The ongoing clinical trials show that **erdafitinib** demonstrated anticancer activity against other cancers, including cholangiocarcinoma, liver cancer, non-small cell lung cancer, prostate cancer, lymphoma and esophageal cancer [78]. The next drug approved by the FDA in 2019 is **pexidartinib**, which is used in the therapy of symptomatic tenosynovial giant cell tumor (TGCT). The drug is a selective inhibitor of the colony-stimulating factor 1 (CSF1) receptor, mast/stem cell growth factor receptor (c-Kit or CD117) and FMS-like tyrosine kinase 3 harboring an internal tandem duplication mutation (FLT3-ITD). The action mechanism of **pexidartinib** is to arrest the kinase in the autoinhibited state by interacting with the CSF1R juxtamembrane region, which prevents an ATP and substrate binding [79].

The multi-kinase inhibitors that already obtained approval for the treatment of metastatic melanoma, an aggressive form of skin cancer with a high mortality rate, are second-generation serine/threonine-protein kinase B-Raf (B-Raf) inhibitors, such as **vemurafenib**, **dabrafenib** or **encorafenib** and mitogen-activated protein kinase (MAPK) kinase (MEK) inhibitors, such as **trametinib**, **cobimetinib** or **binimetinib**. They are the most promising treatment strategies for melanoma consisting of selective inhibition of the active conformation of the B-Raf, especially with V600E mutation [80]. Furthermore, **dabrafenib** and **encorafenib** are also used in the therapy of cancers with several other mutated forms of B-Raf, e.g., V600K-mutated melanomas and V600K/D-mutated melanoma, respectively. The B-Raf inhibitors are characterized by high response rates, a mild, manageable toxicity profile and improved progression-free survival (PFS) as compared with chemotherapy, but their use is limited by the rapid development of resistance [81]. **Encorafenib** is distinguished among the other second-generation B-Raf kinase inhibitors by increasing its inhibitory effect with a shorter off-rate [82]. Currently, monotherapy with B-Raf inhibitor for the treatment of BRAF-mutated melanoma is subsequently replaced by combination therapy with B-Raf and MEK inhibitors, which target key enzymes in the MAPK signaling pathway (RAS-RAF-MEK-ERK). The approved combination of active anticancer ingredients, such as **vemurafenib** plus **cobimetinib**, **dabrafenib** plus **trametinib** or **encorafenib** plus **binimetinib**, is a more effective therapy than B-Raf inhibitor monotherapy and is recommended as a first-line therapeutic option in treating melanoma [83]. What is more, **encorafenib** and **binimetinib** combination therapy is already ongoing clinical development for the treatment of colorectal cancer (CRC). **Selumetinib**, the next inhibitor of MEK 1 and 2, is approved in children with neurofibromatosis type 1 and inoperable plexiform neurofibromas. The long-term treatment with **selumetinib** has meaningful benefits, such as a high level of clinical response and absence of cumulative toxic effects [84].

In 2020, the combination of **encorafenib** and monoclonal antibody (**cetuximab**) received its first approval for treating metastatic colorectal cancer (mCRC) [85]. CRC can also be treated with **regorafenib**, which is an inhibitor of VEGFR-1, VEGFR-2 and VEGFR-3, tunica interna endothelial cell kinase 2 (TIE2), PDGFRB, c-Kit, FGFR1, RET, RAF proto-oncogene serine/threonine-protein kinase (RAF-1) and B-Raf, including wild-type B-Raf and B-Raf V600E [86]. The inhibitor possesses antiangiogenic activity due to the inhibition of TIE2. Moreover, it has anticancer activity against gastrointestinal stromal tumor (GIST), hepatocellular carcinoma and is ongoing clinical development for various malignant tumors. Another two drugs, which were approved in 2020 for GIST, are **avapritinib** and **ripretinib**. They inhibit mast/stem cell growth factor receptor (c-Kit) and platelet-derived growth factor receptor α (PDGFRA). **Avapritinib** is a therapy only for GIST harboring a PDGFRA exon 18 mutations, including PDGFRA D842V mutations, whereas ripretinib inhibits wild-type c-Kit and PDGFRA mutations, as well as multiple primary and secondary resistance mutations in GIST [87]. **Ripretinib** is an appropriate treatment for patients who were resistant to other approved tyrosine kinase inhibitors, such as **regorafenib** or **imatinib**. The mechanism of its action involves durably binding to both the switch pocket in the intracellular juxtamembrane domain and the activation loop in the kinase domain to prevent from adopting an active state of kinase and locking it in the inactive conformation, thereby inhibiting cell proliferation [88].

A total of six drugs, which have been approved by the FDA since 2011 for thyroid cancer treatment, are antiangiogenic multi-kinase inhibitors, including **vandetanib**, **cabozantinib** and **lenvatinib** or mutation-specific inhibitors, including **dabrafenib** for BRAF-mutated anaplastic thyroid cancer (ATC), **larotrectinib** for NTRK-fusion thyroid cancer and **selpercatinib** for RET-mutant medullary thyroid cancer (MTC). **Vandetanib** and **cabozantinib** are registered for the treatment of advanced MTC. The drugs inhibit EGF, RET and VEGF receptors or the MET, RET and VEGF receptors, respectively. Thus, their action involves blocking the sustaining proliferative signaling mediated by tyrosine kinase receptors, angiogenesis and apoptosis. **Cabozantinib**, due to downregulation of the MET pathway, may prevent invasiveness and metastatic spread of cancer cells and the development of acquired resistance. Hence, it induces more prolonged clinical responses than those to other TKIs. What is more, it displays stronger antiangiogenic activity than **vandetanib** [89]. However, the use of **cabozantinib** and **vandetanib** is at least partially limited by their adverse events. In contrast, next-generation drug **selpercatinib**, which is a highly potent and selective RET inhibitor, shows durable efficacy with a more satisfactory safety profile [90]. The first FDA-approved treatment for patients with anaplastic thyroid carcinomas (ATCs), a highly aggressive and undifferentiated cancer, is **dabrafenib** (B-Raf inhibitor) plus **trametinib** (MEK inhibitor). This dual inhibition improves overall response frequency and achieves better clinical results compared with B-Raf inhibitor monotherapy [91].

**Larotrectinib** is a highly selective TRK inhibitor that was developed for the therapy for cancers with a neurotrophic receptor tyrosine kinase (NTRK) gene fusion in adults and children [92]. All patients undergoing the **larotrectinib** treatment were characterized by advanced solid tumors, including salivary gland tumors, infantile fibrosarcoma, thyroid cancer, NSCLC and other cancers. **Larotrectinib** also has potential efficacy against CNS tumors because of its ability to cross the blood–brain barrier [93]. The only other registered TRK inhibitor apart from **larotrectinib** is **entrectinib**, with activity against ALK, TRK and ROS1. In clinical trials, responses to **entrectinib** treatment were observed in the following diseases: NSCLC, mammary analog secretory carcinoma (MASC), colorectal cancer, melanoma, glioneuronal tumor and renal cell carcinoma (RCC) [94]. **Lenvatinib** is also used in therapy for RCC, as well as radioiodine-refractory differentiated thyroid cancer (RR-DTC), hepatocellular carcinoma and endometrial cancer. The inhibitor targets VEGFR-1, VEGFR-2, VEGFR-3, FGFR1-3, RET, mast/stem cell growth factor receptor (c-Kit) and platelet-derived growth factor receptor β (PDGFRB), thereby resulting in broad spectrum of direct antitumor activity and significant antiangiogenic effects [95]. Another drug for RCC is **axitinib**, which is approved for monotherapy in the second-line treatment and in combination with **pembrolizumab** or **avelumab** for first-line therapy [96,97]. It inhibits both proliferation and angiogenesis through blocking receptors, such as c-Kit and PDGFR on the one hand, and proangiogenic receptors VEGFR-1, VEGFR-2 and VEGFR-3 on the other. **Axitinib** has demonstrated promising activity in other solid tumors as well, including metastatic breast cancer, advanced NSCLC, pancreatic and thyroid cancers [98].

**Table 3 molecules-27-02259-t003:** Features of the multi-kinase inhibitors approved as drugs by the Food and Drug Administration (FDA) from 2011 to 2022. The order of drugs is tabulated in order of most recent to oldest registration date. A generic name of a drug is an international nonproprietary name (INN).

No.	Generic Name of Drug	Brand Name and Company	First FDA/EMA Approval Date	Structure	Molecular Target	Route of Administration	Indication	Adverse Effects	Ref.
1	Ripretinib	QINLOCK Deciphera Pharmaceuticals, Inc., Waltham, MA, USA	FDA: 15 May 2020 EMA: 18 November 2021	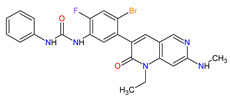	c-Kit ^1^, PDGFRA ^2^	Oral	Gastrointestinal Stromal Tumor	Alopecia, fatigue, nausea, abdominal pain, constipation, myalgia, diarrhea, decreased appetite, palmar–plantar erythrodysesthesia syndrome, vomiting	[99,100]
2	Selpercatinib	RETEVMO Eli Lilly and Company, Indianapolis, IN, USA	FDA: 8 May 2020 EMA: 11 February 2021	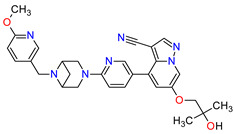	RET ^3^	Oral	Non-Small Cell Lung Cancer, Thyroid Cancer	Increased AST levels, increased glucose levels, decreased albumin levels, decreased leukocyte levels, decreased calcium levels, increased creatinine levels, dry mouth, diarrhea, increased alkaline phosphatase levels, hypertension, fatigue, decreased platelet levels, edema, increased total cholesterol levels, decreased sodium levels, rash, constipation, decreased magnesium levels, increased potassium levels, increased bilirubin levels, headache, decreased glucose levels, nausea, abdominal pain, cough, prolonged QT interval, dyspnea, vomiting, hemorrhage	[101,102]
3	Selumetinib	KOSELUGO AstraZeneca, Cambridge, UK	FDA: 13 April 2020 EMA: 17 June 2021	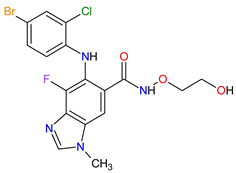	MEK1 ^4^, MEK2 ^5^	Oral	Neurofibromatosis Type 1	Vomiting, rash, abdominal pain, diarrhea, nausea, dry skin, musculoskeletal pain, fatigue, pyrexia, stomatitis, acneiform rash, headache, paronychia, pruritus, dermatitis, constipation, hair changes, epistaxis, hematuria, proteinuria, decreased appetite, decreased cardiac ejection fraction, edema, sinus tachycardia, skin infection	[103,104]
4	Avapritinib	AYVAKIT Blueprint Medicines Corporation, Cambridge, MA, USA	FDA: 9 January 2020 EMA: 24 September 2020	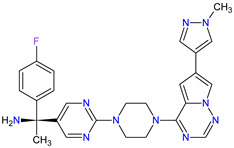	c-Kit ^1^, PDGFRA ^2^	Oral	Gastrointestinal Stromal Tumor	Edema, nausea, fatigue/asthenia, cognitive impairment, vomiting, decreased appetite, diarrhea, increased lacrimation, abdominal pain	[105,106]
5	Entrectinib	ROZLYTREK Genentech, Inc., South San Francisco, CA, USA	FDA: 15 August 2019 EMA: 31 July 2020	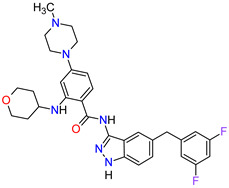	TRK ^6^, ROS1 ^7^, ALK ^8^	Oral	Solid Tumors, Non-Small Cell Lung Cancer	Dysgeusia, fatigue, dizziness, constipation, nausea, diarrhea, increased weight, paresthesia, increased blood creatinine, myalgia, peripheral edema, vomiting, anemia, arthralgia, increased aspartate aminotransferase (AST)	[107,108]
6	Pexidartinib	TURALIO Daiichi Sankyo, Tokyo, Japan	FDA: 2 August 2019 EMA: Not approved	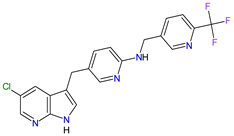	CSF1R ^9^, c-Kit ^1^, FLT3 ^10^	Oral	Tenosynovial Giant Cell Tumor	Hair color changes (depigmentation), fatigue, increased AST, increased alanine aminotransferase (ALT), dysgeusia, vomiting, periorbital edema, abdominal pain, decreased appetite, pruritus, hypertension, increased alkaline phosphatase	[109,110]
7	Erdafitinib	BALVERSA Janssen Pharmaceuticals, Inc., Raritan (HQ), NJ, USA	FDA: 12 April 2019 EMA: Not approved	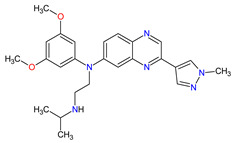	FGFRs ^11^ (1, 2, 3, 4)	Oral	Urothelial Carcinoma	Increased phosphate levels, stomatitis, fatigue, diarrhea, dry mouth, onycholysis, decreased appetite, dysgeusia, dry skin, dry eye, alopecia, palmar–plantar erythrodysaesthesia syndrome, constipation, abdominal pain, nausea, musculoskeletal pain	[78]
8	Larotrectinib	VITRAKVI Loxo Oncology, Inc., Stamford, CT, USA	FDA: 26 November 2018 EMA: 19 September 2019	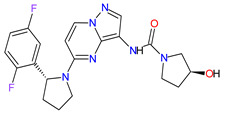	TRK ^6^	Oral	TRK Fusion Cancers	Fatigue, nausea, dizziness, vomiting, anemia, increased transaminase levels, cough, constipation, diarrhea	[111,112]
9	Lorlatinib	LORBRENA Pfizer Inc., New York City, NY, USA	FDA: 2 November 2018 EMA: 6 May 2019	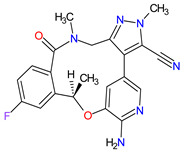	ALK ^8^, ROS1 ^7^	Oral	Non-Small Cell Lung Cancer	Hypercholesterolemia, hypertriglyceridemia, edema, peripheral neuropathy	[113,114]
10	Dacomitinib	VIZIMPRO Pfizer Inc., New York City, NY, USA	FDA: 27 September 2018 EMA: 2 April 2019	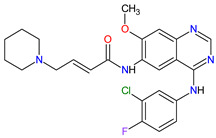	EGFR ^12^, HER2 ^13^, HER4 ^14^	Oral	Non-Small Cell Lung Cancer	Diarrhea, paronychia, dermatitis acneiform, stomatitis, decreased appetite	[115,116]
11	Encorafenib	BRAFTOVI Pfizer Inc., New York City, NY, USA	FDA: 27 June 2018 EMA: 20 September 2018	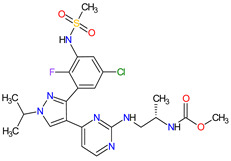	B-Raf ^15^	Oral	Melanoma Metastatic, Colorectal Cancer	Nausea, diarrhea, vomiting, fatigue, arthralgia	[117,118]
12	Binimetinib	MEKTOVI Array BioPharma Inc., Boulder, CO, USA	FDA: 27 June 2018 EMA: 20 September 2018	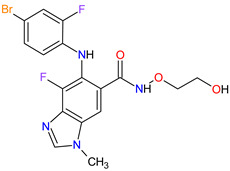	MEK1 ^4^, MEK2 ^5^	Oral	Melanoma Metastatic	Nausea, diarrhea, vomiting, fatigue, arthralgia	[117,119]
13	Brigatinib	ALUNBRIG Takeda Pharmaceuticals America, Inc., Deerfield, IL, USA	FDA: 28 April 2017 EMA: 22 November 2018	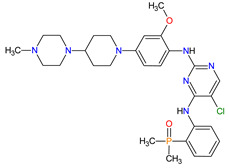	ALK ^8^, EGFR ^12^	Oral	Non-Small Cell Lung Cancer	Nausea, diarrhea, fatigue, cough, headache, CPK elevation, pancreatic enzyme elevation, hyperglycemia	[120,121]
14	Alectinib	ALECENSA Genentech, Inc., South San Francisco, CA, USA	FDA: 11 December 2015 EMA: 16 February 2017	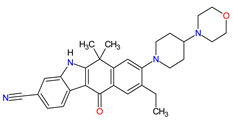	ALK ^8^	Oral	Non-Small Cell Lung Cancer	Constipation, nausea, diarrhea, vomiting, edema, increased levels of bilirubin, AST and ALT, myalgia, rash, anemia, increase in bodyweight	[122,123]
15	Cobimetinib	COTELLIC Genentech, Inc., South San Francisco, CA, USA	FDA: 10 November 2015: EMA: 20 November 2015.	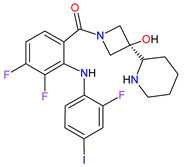	MEK1 ^4^, MEK2 ^5^	Oral	Melanoma Metastatic	Diarrhea, nausea, rash, arthralgia, fatigue, increased creatine phosphokinase levels	[124,125]
16	Lenvatinib	LENVIMAEisai Inc., Tokyo, Japan, U.S. Corporate Headquarters in Nutley, NJ, USA	FDA: 13 February 2015 EMA: 28 May 2015	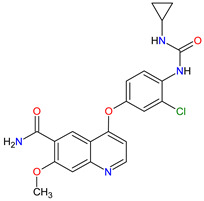	VEGFRs ^16^ (1, 2, 3), FGFR ^11^ (1, 2, 3, 4), PDGFRA ^2^, RET ^3^, c-Kit ^1^	Oral	Thyroid Cancer, Renal Cell Carcinoma, Hepatocellular Carcinoma, Endometrial Cancer	Hypertension, diarrhea, fatigue or asthenia, decreased appetite, bodyweight decreased, nausea, stomatitis, palmar–plantar erythrodysethaesia syndrome, proteinuria	[126,127]
17	Afatinib	GILOTRIF Boehringer Ingelheim Pharmaceuticals, Inc., Ingelheim, Germany	FDA: 12 July 2013 EMA: 25 September 2013	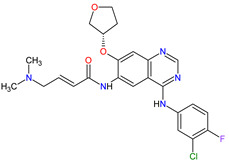	EGFR ^12^, HER2 ^13^, HER4 ^14^	Oral	Non-Small Cell Lung Cancer	Diarrhea, rash/acne, stomatitis/mucositis, paronychia, dry skin, decreased appetite, pruritus, nausea, fatigue, vomiting, epistaxis, cheilitis	[128,129]
18	Trametinib	MEKINIST GlaxoSmithKline, London, UK	FDA: 29 May 2013 EMA: 30 June 2014	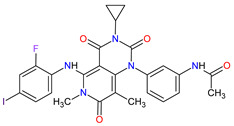	MEK1 ^4^, MEK2 ^5^	Oral	Melanoma, Metastatic, Non-Small Cell Lung Cancer, Thyroid Cancer	Rash, diarrhea, fatigue, nausea/vomiting, peripheral edema	[130,131]
19	Dabrafenib	TAFINLAR GlaxoSmithKline, London, UK	FDA: 29 May 2013 EMA: 26 August 2013	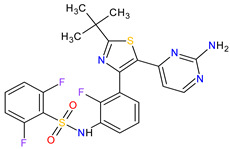	B-Raf ^15^	Oral	Melanoma, Metastatic, Non-Small Cell Lung Cancer, Thyroid Cancer	Alopecia, arthralgia, back pain, constipation, cough, erythrodysaesthesia, fever, headache, hyperkeratosis, muscle pain, nasopharyngitis, papilloma, squamous cell cancer	[132,133]
20	Cabozantinib	CABOMETYX Exelixis, Inc., Alameda, CA, USA	FDA: 25 April 2016 EMA: 9 September 2016	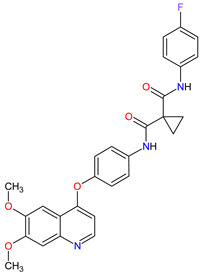	MET ^17^, RET ^3^, VEGFRs ^16^ (1, 2, 3), c-Kit ^1^, FLT-3 ^10^, TIE2 ^18^, TRKB ^19^, AXL ^20^	Oral	Renal Cell Carcinoma, Hepatocellular Carcinoma	Diarrhea, fatigue, nausea, vomiting, decreased appetite, hypertension, palmar–plantar erythrodysesthesia syndrome	[134,135,136]
21	Cabozantinib	COMETRIQ Exelixis, Inc., Alameda, CA, USA	FDA: 29 November 2012 EMA: 21 March 2014	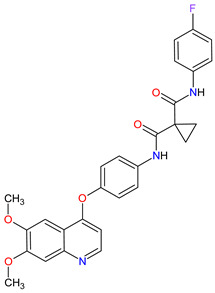	MET ^17^, RET ^3^, VEGFRs ^16^ (1, 2, 3), c-Kit ^1^, FLT-3 ^10^, TIE2 ^18^, TRKB ^19^, AXL ^20^	Oral	Thyroid Cancer	Diarrhea, stomatitis, palmar–plantar erythrodysesthesia syndrome, decreased weight, decreased appetite, nausea, fatigue, oral pain, hair color changes, dysgeusia, hypertension, abdominal pain, constipation, increased AST, increased ALT, lymphopenia, increased alkaline phosphatase, hypocalcemia, neutropenia, thrombocytopenia, hypophosphatemia, and hyperbilirubinemia	[137,138,139]
22	Regorafenib	STIVARGA Bayer HealthCare Pharmaceuticals Inc., Whippany, NJ, USA	FDA: 27 September 2012 EMA: 26 August 2013	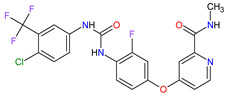	VEGFRs ^16^ (1, 2, 3), RET ^3^, c-Kit ^1^, PDGFRs ^21^ (A, B), FGFRs ^11^ (1, 2), TIE2 ^18^, B-Raf ^15^, RAF-1 ^22^	Oral	Colorectal Cancer, Gastrointestinal Stromal Tumor, Hepatocellular Carcinoma	Asthenia/fatigue, decreased appetite and food intake, hand-foot skin reaction, palmar–plantar erythrodysesthesia, diarrhea, mucositis, weight loss, infection, hypertension, dysphonia	[140,141,142]
23	Axitinib	INLYTA Pfizer Inc., New York City, NY, USA	FDA: 27 January 2012 EMA: 3 September 2012	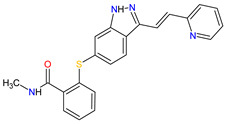	VEGFRs ^16^ (1, 2, 3), c-Kit ^1^, PDGFRs ^21^ (A, B)	Oral	Renal Cell Carcinoma	Diarrhea, hypertension, fatigue, decreased appetite, nausea, dysphonia, palmar–plantar erythrodysesthesia (hand-foot) syndrome, weight decreased, vomiting, asthenia, constipation	[143,144,145]
24	Crizotinib	XALKORI Pfizer Inc., New York City, NY, USA	FDA: 26 August 2011 EMA: 23 October 2012	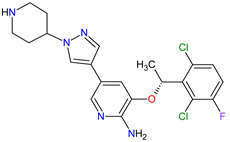	ALK ^8^, MET ^17^, ROS1 ^7^	Oral	Non-Small Cell Lung Cancer	Vision disorders, nausea, diarrhea, vomiting, edema, constipation, elevated transaminases, fatigue, decreased appetite, upper respiratory infection, dizziness, neuropathy	[146,147,148]
25	Vemurafenib	ZELBORAF Genentech, Inc., South San Francisco, CA, USA	FDA: 17 August 2011 EMA: 17 February 2012	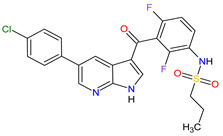	B-Raf ^15^	Oral	Melanoma Metastatic	Arthralgia, rash, alopecia, fatigue, photosensitivity reaction, nausea,	[149,150,151]
26	Vandetanib	CAPRELSA AstraZeneca, Cambridge, UK	FDA: 6 April 2011 EMA: 17 February 2012	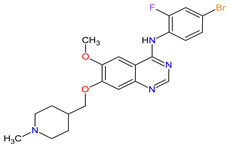	VEGFR-2 ^23^, EGFR ^12^, RET ^3^	Oral	Thyroid Cancer	Diarrhea, rash, nausea, hypertension, fatigue, headache, decreased appetite, acne, dermatitis acneiform, dry skin, photosensitivity reaction, erythema	[152,153]

^1^ **c-Kit**: mast/stem cell growth factor receptor. ^2^ **PDGFRA**: platelet-derived growth factor receptor α. ^3^ **RET**: receptor tyrosine kinase rearranged during transfection. ^4^ **MEK1**: mitogen-activated protein kinase kinase 1. ^5^ **MEK2**: mitogen-activated protein kinase kinase 2. ^6^ **TRK**: tropomyosin receptor tyrosine kinase. ^7^ **ROS1**: proto-oncogene tyrosine-protein kinase ROS. ^8^ **ALK**: anaplastic lymphoma kinase. ^9^ **CSF1R**: colony-stimulating factor 1 receptor. ^10^ **FLT3**: FMS-like tyrosine kinase-3. ^11^ **FGFRs**: fibroblast growth factor receptors. ^12^**EGFR**: epidermal growth factor receptor. ^13^ **HER2**: human epidermal growth factor receptor 2. ^14^ **HER4**: human epidermal growth factor receptor 4. ^15^ **B-Raf**: serine/threonine-protein kinase B-Raf. ^16^ **VEGFRs**: vascular endothelial growth factor receptors. ^17^ **MET**: mesenchymal-epithelial transition factor. ^18^ **TIE2**: tunica interna endothelial cell kinase 2. ^19^ **TRKB**: tropomyosin receptor kinase B. ^20^ **AXL**: AXL receptor tyrosine kinase. ^21^ **PDGFRs**: platelet-derived growth factor receptors. ^22^ **RAF-1**: RAF proto-oncogene serine/threonine-protein kinase. ^23^ **VEGFR-2**: vascular endothelial growth factor receptor-2.

## 3. Phosphatidylinositol 3-Kinase α Inhibitors as Anticancer Agents

Phosphatidylinositol 3-kinase α (PI3K-α) is a lipid kinase that catalyzes the phosphorylation reaction of phosphoinositides leading to the generation of phosphatidylinositol 3,4,5-trisphosphate [154]. PI3K-α transmits the extracellular signal by targeting the AKT/mTOR pathway. The deregulation of this pathway is implicated in the progression of solid cancers, particularly those harboring mutation in PIK3CA gene, such as ovarian, colon and breast cancers [155]. PI3K-α selective inhibition, approximately 50 times stronger than against other isoforms, was achieved with **alpelisib**. This drug prevents hyperactivation of the PI3K-α resulting in blocking of the phosphorylation of PI3K downstream targets [156]. **Alpelisib** in clinical trials demonstrated a favorable safety profile and antitumor activity in patients with PIK3CA-altered advanced solid cancers, including PIK3CA-mutant breast cancer [157]. The approval of **alpelisib** (as PIQRAY) was obtained for usage in combination with an estrogen receptor antagonist (**fulvestrant**) for the treatment of postmenopausal women and men with HR-positive, HER2-negative, advanced or metastatic breast cancer [158]. It is used for patients with progression during or after endocrine therapy, which is a standard treatment for breast cancer [156]. The overall features of **alpelisib** are given in Table 4, and its mechanism of action is presented in Figure 4.

## 4. KRAS Inhibitors as Anticancer Agents

GTPase KRas (KRAS) is the protein that transduces activating signals to various cellular signaling pathways, including the regulation of cell proliferation and differentiation. The inactive protein is bound by a guanosine diphosphate (GDP) molecule and is changed to an active guanosine triphosphate (GTP)-bound state when a guanosine exchange factor (GEF) protein displaces GDP from the nucleotide-binding site. This process is critical to facilitate conformational changes in KRAS that allow the protein to bind and activate downstream effector molecules, such as RAF-kinases, PI3K and RalGDS [162]. KRAS mutations present in cancer cells impair hydrolysis of GTP, which results in holding KRAS in a permanently active form. The common but previously elusive alternation is the KRAS^G12C^ mutation characterized with substitution of a glycine amino-acid residue for a cysteine residue, which occurs most frequently in lung cancer [163]. The discovery of **sotorasib** and its approval by the FDA (as LUMAKRAS), a first-in-class inhibitor that selectively targets KRAS^G12C^, is a breakthrough advance. The approval of this drug includes the use in adult patients with KRAS G12C-mutated locally advanced or metastatic non-small cell lung cancer (NSCLC) who received treatment of at least one prior systemic therapy [164]. **Sotorasib** binds covalently to the mutant cysteine residue and traps KRAS^G12C^ in its inactive GDP-bound state, thereby inhibiting its downstream signaling effects. The cysteine residue in a pocket of the switch II region (SIIP), which the drug targets, is not present in the wild-type or other mutant KRAS that allows avoiding off-target action [165]. **Sotorasib** can exist in either of two atropisomeric forms. The (*R*)-atropisomer shows substantial occupancy in the S-IIP pocket and is a more active KRAS inhibitor compared to the (*S*)-atropisomer [166]. The characteristics of **sotorasib** are summarized in Table 5 and the schematic representation of its mode of action in Figure 5.

## 5. Various Enzymes Inhibitors as Anticancer Agents

Enzymes are a class of high-molecular-weight proteins responsible for carrying out various chemical reactions of specific enzyme substrates. Enzymes are ideal targets for drugs because drugs can inhibit their activity by attaching to them as false substrates, disrupting the formation of the enzyme-true substrate complex, thereby stopping the enzyme catalyzed reaction from progressing inside the cell. Enzyme inhibitors, depending on their mechanism of action, can be classified as irreversible, competitive and non-competitive [169]. The drugs, which received FDA approval for cancer treatment since 2011, inhibit the catalytic activity of such enzymes as steroid 17alpha-monooxygenase (CYP17A1), poly (ADP-ribose) polymerases (PARPs), thymidylate synthase (TS), topoisomerase (TOP) and enhancer of zeste homolog 2 (EZH2) (Figure 6). The approval characteristics of new enzyme inhibitors are presented in Table 6.

### 5.1. Steroid 17alpha-Monooxygenase Inhibitors

The steroid 17alpha-monooxygenase (CYP17A1) belongs to the family of the cytochrome P450 that catalyzes the conversion of 17-hydroxypregnenolone to dehydroepiandrosterone (DHEA), the main steroid intermediates involved in testosterone synthesis. **Abiraterone acetate** (AA) is a first-in-class CYP17A1 enzyme inhibitor that acts on androgen biosynthesis. It is a prodrug that is converted in vivo into free **abiraterone,** which is the steroidal progesterone derivative [170]. **Abirateone** potently and irreversibly blocks the CYP17A1 enzyme through a covalent binding mechanism. In contrast to **abiraterone**, its acetate ester form is orally bioavailable and is resistant to hydrolysis by esterases. The administration of **abiraterone acetate** lowers the amount of testosterone to castrate-range levels. It had its first approval already in 2011, in combination with prednisone for late-stage castrate-resistant prostate cancer (CRPC) in patients who previously received **docetaxel** chemotherapy, but FDA only granted its full approval on 10 December 2012 [171]. The indication was subsequently extended on 7 February 2018 to include patients with an earlier form of metastatic prostate cancer [170]. **Abiraterone acetate** plus prednisone was also evaluated in phase II trial for the treatment of metastatic or locally advanced breast cancer, where a benefit for some patients with molecular apocrine tumors was observed [172].

### 5.2. Poly (ADP-Ribose) Polymerases Inhibitors

Poly (ADP-ribose) polymerases (PARPs) are a family of enzymes that catalyze the transfer of ADP-ribose to target proteins. There are 18 proteins that had been identified as members of the PARPs family. PARPs regulate important cellular processes, including modulation of chromatin structure, nucleic acid metabolism, DNA synthesis and DNA repair. The cancer cells use PARP-mediated DNA repair for homologous recombination and survival, thereby being sensitive to PARPs inhibition [173]. In the last decade, **olaparib**, **rucaparib**, **niraparib** and **talazoparib** have been FDA-approved inhibitors of PARPs catalytic activity. The drugs cause the blocking repair of DNA single-strand breaks (SSBs), which allow progressing DNA damage to double-stranded breaks (DSBs), as well as trap the PARP1 and PARP2 enzymes at damaged DNA that induces synthetic lethality of cancer cells. PARP inhibitors may also increase cancer sensitivity to DNA-damaging agents [174]. The first indication that **olaparib**, which was FDA approved, is BRCA mutation-positive advanced epithelial ovarian cancer (EOC), fallopian tube, or primary peritoneal carcinoma. Following the approval of **olaparib**, **rucaparib** and **niraparib** were also registered for the treatment of relapsed BRCA-mutated ovarian cancer. In 2018, **olaparib** and **talazoparib** obtained authorization for patients with germline BRCA-mutated (gBRCAm), HER2-negative metastatic breast cancer who were treated with chemotherapy [175]. A year later, i.e., in 2019, the FDA accepted the New Drug Application for **olaparib** for the treatment of deleterious or suspected deleterious gBRCAm metastatic pancreatic adenocarcinoma [176]. Finally, in 2020, both **olaparib** and **rucaparib** were approved for the therapy of patients with metastatic castration-resistant prostate cancer (mCRPC) [177].

### 5.3. Topoisomerase I Inhibitors

Topoisomerases induce transient DNA breaks allowing relaxing the supercoiled DNA and carrying out catalytic functions during transcription and replication. There are six human topoisomerases that act on a broad range of DNA and RNA substrates at the nuclei and mitochondria. They can be dived into two classes: type I, including type IA (TOP3α and TOP3β) and IB (TOP1 and TOP1mt) that cleave one strand of DNA and type II (TOP2α and TOP2β) that cleave both strands [178]. The semisynthetic derivatives of camptothecin, such as **irinotecan**, which was first FDA approved in 1996 for patients with metastatic colorectal cancer (CRC) [179], are a class of anticancer drugs acting through reversible inhibition of topoisomerase I (TOP1). Their lactone form binds to the TOP1 cleavable complex, traps the enzyme on DNA, resulting in inhibition of replication of the single-strand DNA and induction of irreversible double-strand breaks. These events lead to the arrest of the cell cycle in the S-G2 phase and cell death. Irinotecan is a prodrug that exerts its cytotoxic effect due to the rapid metabolic formation of highly potent 7-ethyl-10-hydroxycamptothecin (SN-38) [180]. The use of the drug was enriched with FDA approval in 2015 of **irinotecan** in a nanoliposomal formulation (ONIVYDE, Merrimack Pharmaceuticals, Inc.). This new liposomal form of **irinotecan** improves the pharmacokinetics and biodistribution of the drug, as well as protects irinotecan from premature metabolism and interconversion of its active lactone form to the inactive carboxylate. Moreover, it prolongs the circulation time of the drug, thereby extending its exposure at the site of action and enhancing its efficacy [181]. A liposomal version of **irinotecan** in combination with **leucovorin**-modulated **fluorouracil** was registered for the treatment of metastatic pancreatic adenocarcinoma. This therapy significantly extends the survival of patients without deteriorating their quality of life [182].

### 5.4. Enhancer of Zeste Homolog 2 Inhibitors

The enhancer of zeste homolog 2 (EZH2) is a histone methyltransferase that catalyzes trimethylation of histone H3 at Lys 27 (H3K27me3). EZH2 is also a catalytic component of polycomb repressive complex 2 (PRC2), which is a group of important epigenetic regulators of gene expression for regulating the differentiation of healthy cells. Mutation or overexpression of the EZH2 gene plays a critical role in the development of various cancers, such as CRC, melanoma, ovarian cancer and breast cancer. Dysregulation of EZH2 as a histone modifier causes proliferation of cancer cells and promotes their survival and metastasis, resulting in invasion and progression of a malignant tumor. Moreover, EZH2 is involved in the regulation of immune cells (e.g., T cells, NK cells, dendritic cells and macrophages), which are essential components in the cancer microenvironment [183]. **Tazemetostat** is a first-in-class, potent and highly selective EZH2 inhibitor. The mechanism of its action is blocking EZH2 activity, thus the trimethylation of H3K27 that results in tumor regressions. In clinical development, **tazemetostat** showed a favorable safety profile and responses in patients with either lymphoma, including both germinal center B-cell like (GCB) and non-GCB subtypes of diffuse large B-cell lymphoma (DLBCL) [184]. The drug was originally approved by the FDA as the first therapy for the treatment of adults and adolescents aged 16 years and older with locally advanced or metastatic epithelioid sarcoma not eligible for complete resection [185].

**Table 6 molecules-27-02259-t006:** Features of the various enzymes inhibitors approved as drugs by the Food and Drug Administration (FDA) from 2011 to 2022. The order of drugs is tabulated in order of most recent to oldest registration date. A generic name of a drug is an international nonproprietary name (INN).

No.	Generic Name of Drug	Brand Name and Company	First FDA/EMA Approval Date	Structure	Molecular Target	Route of Administration	Indication	Adverse Effects	Ref.
1	Tazemetostat	TAZVERIK Epizyme, Inc., Cambridge, MA, USA	FDA: 23 January 2020 EMA: Not approved	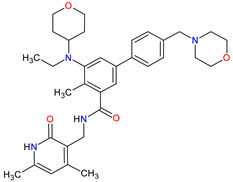	EZH2 ^1^	Oral	Epithelioid Sarcoma	Pain, fatigue, nausea, decreased appetite, vomiting, constipation	[186]
2	Talazoparib	TALZENNA Pfizer Inc., New York City, NY, USA	FDA: 16 October 2018 EMA: 20 June 2019	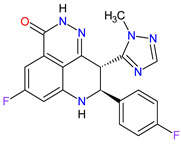	PARPs ^2^ (1, 2)	Oral	Breast Cancer	Anemia	[187,188]
3	Niraparib	ZEJULA GlaxoSmithKline, London, UK	FDA: 27 March 2017 EMA: 16 November 2017	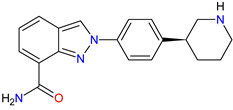	PARPs ^2^ (1, 2)	Oral	Ovarian Cancer, Fallopian Tube Cancer, Peritoneal Cancer	Hematological abnormalities, palpitations, gastrointestinal events, mucositis/stomatitis, dry mouth, fatigue/asthenia, urinary tract infection, aminotransferase enzyme elevations, myalgia, back pain, arthralgia, headache, dizziness, dysgeusia, insomnia, anxiety, nasopharyngitis, dyspnea, cough, rash, hypertension	[189,190]
4	Rucaparib (as camsylate)	RUBRACA Clovis Oncology, Inc., Boulder, CO, USA	FDA: 19 December 2016 EMA: 24 May 2018	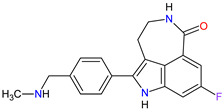	PARPs ^2^ (1, 2, 3)	Oral	Ovarian Cancer, Prostate Cancer	Nausea, fatigue (including asthenia), vomiting, anemia, abdominal pain, dysgeusia, constipation, decreased appetite, diarrhea, thrombocytopenia, dyspnea, increase in creatinine, alanine aminotransferase (ALT), aspartate aminotransferase (AST) and cholesterol, decrease in hemoglobin, lymphocytes, platelets and neutrophils	[191,192]
5	Irinotecan	ONIVYDE Merrimack Pharmaceuticals, Inc., Cambridge, MA, USA	FDA: 22 October 2015 EMA: 14 October 2016	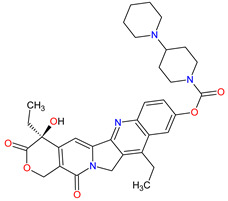	TOP1 ^3^	Injection	Pancreatic Cancer	Diarrhea, fatigue, vomiting, nausea, decreased appetite, stomatitis, fever, lymphopenia, neutropenia	[193,194,195]
6	Olaparib	LYNPARZA AstraZeneca, Cambridge, Great Britain	FDA: 19 December 2014 EMA: 16 December 2014.	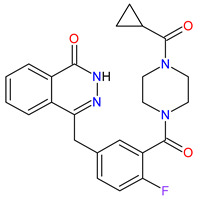	PARPs ^2^ (1, 2)	Oral	Ovarian Cancer, Fallopian Tube Cancer, Peritoneal Cancer, Breast Cancer, Pancreatic Cancer, Prostate Cancer	Nausea, fatigue, vomiting, anemia	[196,197,198]
7	Abiraterone acetate	ZYTIGA Janssen Biotech, Inc., Horsham, PA, USA	FDA: 28 April 2011 EMA: 5 September 2011	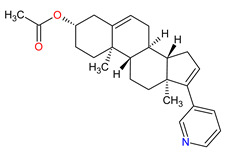	CYP17A1 ^4^	Oral	Prostate Cancer	Joint swelling or discomfort, hypokalemia, edema, muscle discomfort, hot flush, diarrhea, urinary tract infection, cough, hypertension, arrhythmia, urinary frequency, nocturia, dyspepsia, upper respiratory tract infection	[199,200,201]

^1^ **EZH2**: enhancer of zeste homolog 2. ^2^ **PARPs**: poly (ADP-ribose) polymerases. ^3^ **TOP1**: topoisomerase 1. ^4^ **CYP17A1**: steroid 17alpha-monooxygenase.

## 6. Various Receptors Antagonists as Anticancer Agents

Receptors are chemical structures composed of a special class of proteins that has an affinity for specific ligand molecules. The receptor change conformation after binding of a ligand results in some form of cellular response, including relaying a signal into the cell [202]. The deregulation of various receptors’ activity is frequent in solid cancers and may be a molecular target for the development of new anti-cancer agents. The receptor antagonists approved by the FDA since 2011 target such receptors as the smoothened (SMO) receptor, androgen receptor (AR) and pituitary gonadotropin-releasing hormone receptor (GnRH-r) (Figure 7). All new drugs are characterized in Table 7.

### 6.1. Smoothed Receptor Antagonists

The smoothened (SMO) receptor belongs to a G-protein-coupled receptor (GPCR) family. SMO transduces signal in the hedgehog (Hh) pathway, which plays a crucial role in normal embryonic development and maintenance or repair of mature tissue [205]. Under physiological conditions, the activation of the SMO receptor is regulated by the binding of hedgehog signaling proteins to a patched (PTCH) receptor, which results in the downstream activation of Hh signaling cascade. The abnormal Hh signaling is involved in solid cancer formation, such as basal cell carcinoma (BCC) or medulloblastoma, and supports the tumor microenvironment in disparate cancers [206]. The SMO receptor activity can be modulated by small-molecule inhibitors, some of which are currently FDA-approved anticancer drugs, including **vismodegib**, **sonidegib** and **glasdegib**. **Vismodegib** and **sonidegib** are selective and potent SMO receptor antagonists for the treatment of unresectable advanced or locally advanced basal cell carcinoma, respectively [207,208]. Both agents show the same efficacy and safety profiles, but their pharmacokinetic profiles revealed several differences, such as the volume of distribution and half-life [209]. The drugs disrupt aberrant activation of Hh pathway signaling due to inhibition of SMO through interaction with its drug-binding pocket. Additionally, those inhibitors are under intensive investigation in ongoing clinical trials for therapy of other cancer types, such as medulloblastoma, multiple myeloma, myelofibrosis, pancreatic cancer, prostate cancer, breast cancer, ovarian cancer, chronic myeloid leukemia, myelodysplastic syndromes, esophageal cancer and chondrosarcoma. However, as with all chemotherapeutic drugs, **vismodegib** and **sonidegib** are prone to evolution drug resistance, especially by acquired mutations in the SMO receptor.

### 6.2. Androgen Receptor Antagonists

Androgen receptor (AR) belongs to the steroid hormone receptor superfamily. AR is a ligand-dependent transcription factor that regulates androgen-responsive genes involved in various physiological processes, particularly male sexual differentiation and maturation [210]. AR binds to specific DNA sequences known as androgen-responsive elements that activate the androgen-induced transcription of target genes [211]. The activation of AR is also modulated by the interaction of AR with other transcription factors or co-regulatory proteins and by phosphorylation of AR [210]. The overexpression of the AR has been found responsible for the failure of endocrine therapy in some prostate cancers, including castration-resistant prostate cancer (CRPC) or metastatic castration-sensitive prostate cancer (mCSPC) [212]. Treatment options for CRPC and mCSPC resulting in increased patient lifespan and extended metastasis-free overall survival (OS) include the second-generation AR inhibitors, i.e., **enzalutamide**, **apalutamide** and **darolutamide**. The first approval of **enzalutamide** was obtained for the treatment of patients with metastatic CRPC who have previously been treated with **docetaxel**. In 2018, the FDA registered the new indication for **enzalutamide** and the first approval of **apalutamide** for therapy of patients with non-metastatic CRPC [213,214]. In 2019, the indication of both drugs was extended to include patients with mCSPC, and **darolutamide** received its first approval for treatment of men with non-metastatic CRPC [215]. Although **darolutamide** is structurally distinct from **enzalutamide** and **apalutamide**, all agents are potent AR antagonists that competitively bind to its ligand-binding domain causing both inhibition of AR translocation to the cell nucleus and reduction in its transcriptional activity. They show improved specificity to the AR over other steroidal receptors, as well as a high affinity and no known agonistic effects to the AR. Moreover, the drugs demonstrated full antagonist activity against AR mutants that confer resistance to the first generation of AR inhibitors, such as **bicalutamide**, where it arises from AR amplification or point mutations [204]. 

### 6.3. Gonadotropin-Releasing Hormone Receptor Antagonist

The gonadotropin-releasing hormone receptor (GnRH-r) is a member of the G-protein-coupled receptor (GPCR) family, which is located on the gonadotropic cells in the anterior pituitary [216]. Its activation through binding of the gonadotropin-releasing hormone (GnRH) leads to the biosynthesis and secretion of the gonadotropins, including luteinizing hormone (LH) and follicle-stimulating hormone (FSH). The regulation of GnRH-r activity is significant for gonadal steroidogenesis and reproductive function of the gonads [217]. **Relugolix** is the first orally administered GnRH-r antagonist registered for the treatment of advanced prostate cancer. The mechanism of its action is highly selective binding to and blocking the GnRH-r that inhibits the release of both LH and FSH. This process results in a decrease in the levels of female (i.e., estradiol and progesterone) and male (i.e., testosterone) sex hormones [218]. **Relugolix** rapidly and sustainably reduces the testosterone concentration to castration levels, which leads to inhibition of hormone-dependent prostate cancer cell proliferation. Compared to **leuprolide**, a standard drug for achieving androgen deprivation, **relugolix** is superior in the efficacy of suppression of testosterone levels and cardiovascular safety [219].

**Table 7 molecules-27-02259-t007:** Features of the various receptors antagonists approved as drugs by the Food and Drug Administration (FDA) from 2011 to 2022. The order of drugs is tabulated in order of most recent to oldest registration date. A generic name of a drug is an international nonproprietary name (INN).

No.	Generic Name of Drug	Brand Name and Company	First FDA/EMA Approval Date	Structure	Molecular Target	Route of Administration	Indication	Adverse Effects	Ref.
1	Relugolix	ORGOVYX Myovant Sciences, Inc., Brisbane, CA, USA	FDA: 18 December 2020 EMA: Not approved	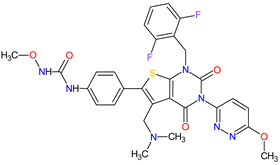	GnRH-r ^1^	Oral	Prostate Cancer	Metrorrhagia, hot flush, viral upper respiratory tract infection, menorrhagia headache, bone density decreased/bone resorption increased	[217]
2	Darolutamide	NUBEQA Bayer HealthCare Pharmaceuticals Inc., Whippany, NJ, USA	FDA: 30 July 2019EMA: 27 March 2020	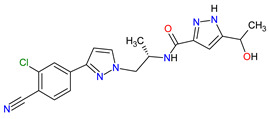	AR ^2^	Oral	Prostate Cancer	Fatigue, extreme pain, rash	[220,221]
3	Apalutamide	ERLEADA Janssen Products, LP, Horsham, PA, USA	FDA: 14 February 2018 EMA: 14 January 2019	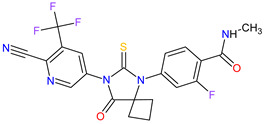	AR ^2^	Oral	Prostate Cancer	Fatigue, high blood pressure, rash, diarrhea, nausea, weight loss, arthralgia, falls, hot flush, decreased appetite, fractures peripheral edema	[222,223]
4	Sonidegib	ODOMZO Novartis Pharmaceuticals Corporation, Basel, Switzerland	FDA: 24 July 2015 EMA: 14 August 2015	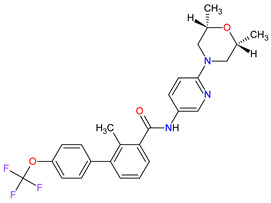	SMO ^3^ receptor	Oral	Basal Cell Carcinoma	Muscle spasms, alopecia, dysgeusia, fatigue, nausea, musculoskeletal pain, diarrhea, decreased weight, decreased appetite, myalgia, abdominal pain, headache, pain, vomiting, pruritus	[224,225]
5	Enzalutamide	XTANDI Astellas Pharma US, Northbrook, IL, USA	FDA: 31 August 2012 EMA: 21 June 2013	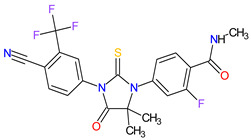	AR ^2^	Oral	Prostate Cancer	Asthenia/fatigue, back pain, diarrhea, arthralgia, hot flush, peripheral edema, musculoskeletal pain, headache, upper respiratory infection, muscular weakness, dizziness, insomnia, lower respiratory infection, spinal cord compression and cauda equina syndrome, hematuria, paresthesia, anxiety, hypertension	[226,227,228]
6	Vismodegib	ERIVEDGE Genentech, Inc., South San Francisco, CA, USA	FDA: 30 January 2012 EMA: 12 July 2013	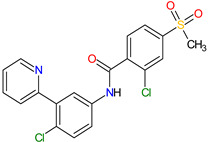	SMO ^3^ receptor	Oral	Basal Cell Carcinoma	Muscle spasms, alopecia, dysgeusia, weight loss, fatigue, nausea, diarrhea, decreased appetite, constipation, arthralgia, vomiting, ageusia	[229,230,231]

^1^ **GnRH-r**: gonadotropin-releasing hormone receptor. ^2^ **AR**: androgen receptor. ^3^ **SMO**: smoothened.

## 7. Transcription Inhibitors as Anticancer Agents

Alkylating agents are the oldest anticancer drugs and remain important for the treatment of several types of cancer. These drugs act by adding an alkyl group to the guanine base of the DNA resulting in breakage of the DNA strands and inhibition of cancer cell proliferation. Since 2011, the FDA approved two alkylating drugs showing the unique mechanism of action, i.e., **trabectedin** and **lurbinectedin**, and JELMYTO, which is a novel formulation of **mitomycin**. An overview of these drugs is presented in Table 8. 

**Trabectedin** is a tetrahydroisoquinoline alkaloid obtained from the marine tunicate *Ecteinascidia turbinate* registered by the FDA in 2015, 8 years after its approval in Europe, for the treatment of specific soft tissue sarcomas (STS)—unresectable or advanced liposarcoma and leiomyosarcoma. **Lurbinectedin** is a **trabectedin** analog approved in 2020 for the treatment of adult patients with metastatic small-cell lung cancer (SCLC) who have experienced disease progression despite therapy with platinum-based agents. The mechanism of action of these drugs is binding to the minor groove of DNA in GC-rich sequences and forming covalent adducts with the amino group of guanine. This consequently leads to termination of the cell cycle in the G2/M phase and ultimately cell death [232,233]. They also interfere with the DNA-repair pathways, resulting in double-strand DNA breaks (DSBs). Moreover, both drugs inhibit the active transcription by direct degradation of the RNA polymerase II during elongation [234,235]. Finally, **trabectedin** and **lurbinectedin** modulate the tumor microenvironment by inducing rapid apoptosis in mononuclear phagocytes and tumor-associated macrophages (TAMs), respectively. Another common effect of both drugs is the reduction in angiogenesis as well as inflammatory chemokines (CCL2 and CXCL8) and VEGF [236,237]. Additionally, **lurbinectedin** improves antitumor immunity and induces immunogenic cell death (ICD) [238]. 

JELMYTO is the new formulation of the cytostatic agent, i.e., mitomycin, in reverse thermal gel (RTGel^TM^) (4 mg **mitomycin** per mL gel) that received FDA approval for treatment of low-grade upper tract urothelial cancer (UTUC). The drug is administered into the renal pelvis and calyces via a ureteral catheter tube [239]. **Mitomycin** (also known as **mitomycin-C**) is an antibiotic isolated from *Streptomyces caespitosus* that was originally approved by the FDA on 29 November 1995 [240]. This compound is a bioreductive alkylating agent selective toward DNA. It induces a variety of cytotoxic effects on cells, including mono or bifunctional alkylation and the cross-linking of DNA [241]. The second component of JELMYTO is a thermosensitive hydrogel, which is in a liquid form at room temperature and transforms into a semisolid gel when exposed to body temperature. The drug can, therefore, be easily administrated in a liquid state and then solidified on the application site [242]. Reverse-thermal hydrogel formulation provides a sustained exposure that allows for increased drug concentration and dwelling time at the site of tumor compared with aqueous solutions. It also optimizes drug delivery and controls its release, leading to a reduction in off-target effects [239].

**Table 8 molecules-27-02259-t008:** Features of the alkylating drugs approved as drugs by the Food and Drug Administration (FDA) from 2011 to 2022. The order of drugs is tabulated in order of most recent to oldest registration date. A generic name of a drug is an international nonproprietary name (INN).

No.	Generic Name of Drug	Brand Name and Company	First FDA/EMA Approval Date	Structure	Molecular Target	Route of Administration	Indication	Adverse Effects	Ref.
1	Lurbinectedin	ZEPZELCA PharmaMar (Colmenar Viejo, Spain) and Jazz Pharmaceuticals plc, Dublin, Ireland	FDA: 15 June 2020 EMA: Not approved	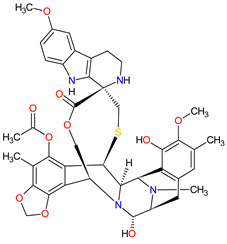	DNA ^1^, RNA ^2^	Injection	Small Cell Lung Cancer	Myelosuppression, fatigue, increased creatinine, increased alanine aminotransferase, increased glucose, nausea, decreased appetite, musculoskeletal pain, decreased albumin, constipation, dyspnea, decreased sodium, increased aspartate aminotransferase, vomiting, cough, decreased magnesium and diarrhea	[243]
2	Mitomycin	JELMYTO UroGen Pharma Ltd., New York, NY, USA	FDA: 15 April 2020 EMA: Not approved	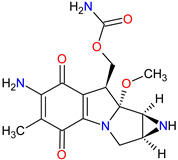	DNA ^1^	Standard Ureteral Catheters	Urothelial Carcinoma	Ureteric obstruction, flank pain, urinary tract infection, hematuria, renal dysfunction, fatigue, nausea, abdominal pain, dysuria, vomiting	[244]
3	Trabectedin	YONDELIS Janssen Biotech, Inc., Horsham, PA, USA	FDA: 23 October 2015 EMA: 17 September 2007	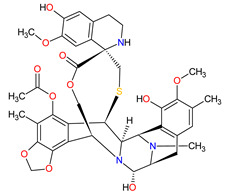	DNA ^1^, RNA ^2^	Injection	Soft Tissue Sarcoma	Nausea, fatigue, vomiting, constipation, decreased appetite, diarrhea, peripheral edema, dyspnea, headache, neutropenia, increased ALT, thrombocytopenia, anemia, increased AST, increased creatine phosphokinase	[245,246,247]

^1^ **DNA**: deoxyribonucleic acid. ^2^ **RNA**: ribonucleic acid.

## 8. Therapeutic Radiopharmaceuticals as Anticancer Agents

Therapeutic radiopharmaceuticals are radiolabeled molecules designed to deliver appropriate radioisotopes to cancer cells, where they bind preferentially or accumulate by physiological mechanisms. Radionuclides are used to directly target highly potent forms of radiation, especially gamma radiation (i.e., γ rays that are high-energy electromagnetic radiation arising from the radioactive decay of atomic nuclei), beta radiation caused by β-particles (i.e., high-energy, high-speed electrons or positrons and alpha radiation caused by α-particles (i.e., helium-4 nuclei) [248]. The carrier molecules for radioactive atoms include antibodies, antibody fragments, proteins, peptides and small molecules. Compared with other methods of treating many types of cancer, therapeutic radiopharmaceuticals have become safe and effective therapy [249]. The features of therapeutic radiopharmaceuticals approved by the FDA since 2011, i.e., **iobenguane I 131**, **lutetium Lu 177 dotatate** and **radium-223 dichloride**, are presented in Table 9.

**Iodine-131 metaiodobenzylguanidine**, known as **iobenguane I 131** (131I-MIBG), is a structural analog of the neurotransmitter norepinephrine radiolabeled with ¹³¹I, which was approved by the FDA in 1994 as a diagnostic agent for the localization of rare tumors of the adrenal gland, such as pheochromocytomas (PHEOs) and paragangliomas (PGLs). The development of the high-specific-activity (HSA) I-131 MIBG at the beginning of the 21st century allowed its FDA approval in 2018 for the treatment of adult and pediatric patients with unresectable PHEO/PGL [250]. HSA-I-131-MIBG is effective as a single agent for patients with locally advanced or metastatic PHEO/PGL treated with a high dose of the drug. At this dose level, the therapy causes enough β (primarily) and γ (secondarily) radiation for the formation of free radicals that damage DNA through single- or double-strand brakes. Therefore, high-dose I-131 MIBG treatment results in better therapeutic responses and higher overall survival (OS) rate compared to low-dose therapy [251].

**Lutetium Lu 177 dotatate** is a radiolabeled somatostatin analog for the treatment of adults with somatostatin receptor (SSTR)-positive gastroenteropancreatic neuroendocrine tumors (GEP-NETs). The drug belongs to the peptide receptor radionuclide therapy (PRRT) class, which was first approved by the FDA, and is composed of the radionuclide ^177^Lu, which is a short-range β-particle emitter, and the bifunctional chelator (1,4,7,10-tetraazacyclododecane-1,4,7,10-tetra-acetic acid, DOTA) that is bound to the peptide (Tyr3)-octreotate. The drug targets cancer cells expressing somatostatin receptors (SSTRs) with high affinity to subtype 2 receptors. The mechanism of its action includes DNA single- and double-strand breaks provoked by β-particle, which in the case of double-strand breaks results in damage of SSTR-positive cells [252].

**Radium-223 dichloride**, which contains the isotope 223 of radium discovered by Godlewski [253,254], is a first-in-class α-particle emitter approved by the FDA for the treatment of metastatic castration-resistant prostate cancer with symptomatic bone metastases, without known visceral metastases disease. The drug has favorable tolerability and is emerging as a valuable option for patients with this poor-prognosis metastatic disease [255]. **Radium-223 dichloride**, where radium atom is a calcium mimetic that is absorbed into the newly formed bone matrix within metastatic lesions. It acts through the induction of irreversible DNA double-strand breaks in the abnormal bones by α particles leading to cancer cell death. Its high linear energy transfer (LET) causes cytotoxic effects that are independent of dose rate, cell cycle status and oxygen concentration, which is especially important in the event of bones being quite hypoxic organs [256].

**Table 9 molecules-27-02259-t009:** Features of the therapeutic radiopharmaceuticals approved as drugs by the Food and Drug Administration (FDA) from 2011 to 2021. The order of drugs is tabulated in order of most recent to oldest registration date. A generic name of a drug is an international nonproprietary name (INN).

No.	Generic Name of Drug	Brand Name and Company	First FDA/EMA Approval Date	Structure	Molecular Target	Route of Administration	Indication	Adverse Effects	Ref.
1	Iobenguane I 131	AZEDRA Progenics Pharmaceuticals, Inc., New York, NY, USA	FDA: 30 July 2018 EMA: Nationally authorized	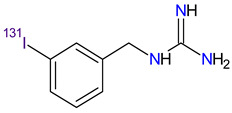	DNA ^1^	Injection	Pheochromocytoma, Paraganglioma	Lymphopenia, neutropenia, thrombocytopenia, fatigue, anemia, increased international normalized ratio, nausea, dizziness, hypertension, vomiting	[257,258]
2	Lutetium Lu 177 dotatate	LUTATHERA Advanced Accelerator Applications S.A., Saint-Genis-Pouilly, France	FDA: 26 January 2018 EMA: 26 September 2017	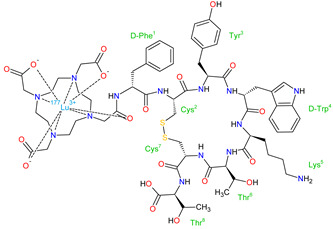	SSTRs ^2^	Injection	Gastroenteropancreatic Neuroendocrine Tumors	Lymphopenia, increased gamma-glutamyltransferase, vomiting, nausea, increased AST, increased ALT, hyperglycemia, hypokalemia	[252,259,260]
3	Radium-223 dichloride	XOFIGO Bayer HealthCare Pharmaceuticals Inc., Whippany, NJ, USA	FDA: 15 May 2013 EMA: 13 November 2013	^223^RaCl_2_	DNA ^1^	Injection	Prostate Cancer	Nausea, diarrhea, vomiting, peripheral edema	[261,262]

^1^ **DNA**: deoxyribonucleic acid. ^2^ **SSTRs**: somatostatin receptors.

## 9. Fixed-Dose Combination Drugs as Anticancer Agents

A fixed-dose combination (FDC) drug consists of two or more active agents within a single form of pharmaceutical administration. For example, LONSURF is a fixed-dose combination of **trifluridine** and **tripacil** in a molar ratio of 1:0.5. (Table 10). Both ingredients in the drug reach a target cancer cell simultaneously and act by different mechanisms, which improve inhibition of growth of solid cancer and, hence, the therapeutic efficacy [263]. **Trifluridine** is an inhibitor of thymidylate synthase (TS), while **tripacil** is an inhibitor of thymidine phosphorylase (TP). **Trifluridine** is a thymidine-based nucleoside analog that was originally approved by the FDA on April 10 1980 as an antiviral drug with the brand name VIROPTIC [264]. **Trifluridine** is converted intracellularly to its active metabolites, i.e., trifluorothymidine monophosphate, which reversibly binds to the active site of TS. The enzyme TS catalyzes the reductive methylation of deoxyuridine monophosphate (dUMP) to deoxythymidine monophosphate (dTMP), which provides the sole de novo source of thymidylate necessary for DNA replication and repair [265]. Subsequent further phosphorylation of trifluorothymidine monophosphate results in the production of trifluridine triphosphate, which is readily incorporated into replicating DNA strands of cancer cells, thereby inhibiting DNA synthesis and ultimately cellular proliferation. **Trifluridine** induces cell death via both caspase-dependent and independent mechanisms. This agent potently induces levels of cell death and does not elicit an autophagic survival response in the cancer cell lines [266]. As mentioned above, it also exhibits antiviral activity [267]. **Tipiracil** is used in LONSURF to increase systemic exposure of **trifluridine** through inhibition of the enzyme thymidine phosphorylase (TP), which is responsible for the breakdown of the active **trifluridine** component. **Tipiracil** component also inhibits angiogenesis in solid cancer and migration of endothelial cells [266]. Overall, LONSURF has two mechanisms of action related to the cytotoxicity of the antimetabolite component **trifluridine** and the antiangiogenic effect of **tipiracil**. LONSURF was first approved for the treatment of adult patients with metastatic colorectal cancer (mCRC) who were refractory to or were not considered candidates for current standard chemotherapy. On 25 February 2019, the use of LONSURF was extended for adult patients with metastatic gastric or gastroesophageal junction (GEJ) adenocarcinoma previously treated with at least two prior lines of chemotherapy or, if appropriate, targeted therapy [268]. 

## 10. Potential Anticancer Drugs in the Pipeline

Before being approved, new anticancer agents undergo multi-phase clinical trials whose purpose is to evaluate their efficacy and safety profile for patients who will take them in clinical practice. The results from these trials are subsequently evaluated by regulatory agencies, including the FDA and the EMA, which decide whether a drug can be registered and, if so, its indications and limitations. The anticancer agents represent a high percentage of all drug candidates, so-called drugs pipeline, which demonstrate the potential to advance the treatment of life-threatening diseases. These small molecule compounds demonstrate promising antitumor activity by targeting various kinases, receptors and proteins. In Table 11, examples are shown of anticancer drugs in the pipeline, including candidates in Phase II to III. It is worth noting that two of the presented molecules, i.e., **adavosertib** (formerly **AZD1775**) and **pemrametostat** (formerly **GSK3326595**), are characterized by a completely new mechanism of action, such as WEE1 G2 checkpoint kinase (WEE1) inhibition and protein arginine methyltransferase 5 (PRMT5) inhibition, respectively [271,272]. The next two agents, i.e., **ARV-471** and **giredestrant** (also known as **GDC-9545**) are specifically designed to target and degrade the estrogen receptor (ER) during the treatment of patients with breast cancer [273,274]. It should be emphasized that **ARV-471** is one of the first clinically evaluated proteolysis-targeting chimera (PROTAC) molecules. This agent consists of two active domains joined by a linker, wherein one domain binds a protein of interest (POI) and the other binds an E3 ubiquitin ligase, resulting in an induction of selective intracellular proteolysis. The usage of PROTAC molecules improves oral exposure, efficacy and safety in the clinic [273]. Since protein kinases are a common molecular target in drug design, there are a lot of protein kinase inhibitors under development and clinical evaluation, for example, AKT inhibitor **ipatasertib**, known as **GDC-0068**, or EGFR and HER2 inhibitor **varlitinib**, known as **ARRY-334543** [275,276]. Finally, a few KRAS inhibitors are in ongoing clinical trials, including **adagrasib** (formerly **MRTX849**), due to the discovery of FDA-approved **sotorasib**, which was the breakthrough advance in developing targeted therapy for patients with non-small cell lung cancer and KRAS mutations [277].

## 11. Conclusions

Considering only low molecular weight compounds, 62 new drug registrations were granted by the FDA from early 2011 up to the end of 2021 (i.e., in the period of eleven years) for use in therapies against solid tumors. A total of 51 of all these drugs were also granted marketing authorization by the EMA (European Medicines Agency). By comparison, in a similar period (between 2011 and 2021), of the 52 drug registrations approved by the FDA for treatment of hematological cancers only, 29 were small molecule compounds [281]. It should be noted that one of the anticancer active compounds, i.e., **cabozantinib**, is commercially available with two registered brand names—COMETRIQ and CABOMETYX. These two drugs are approved in different dosage forms and for different indications, i.e., COMETRIQ as capsules for thyroid cancer and CABOMETYX as tablets for metastatic renal cell carcinoma and hepatocellular carcinoma. Tablets of CABOMETYX did not meet bioequivalence criteria of 80–125% of C_max_ level achieved by capsules of COMETRIQ [282]. Besides, the registered drug with the brand name—LONSURF is composed of two active ingredients, i.e., **trifluridine** and **tipiracil**. On average, four new small molecule drugs were introduced each year, except for 2015, 2018 and 2020, when there were 9, 9 and 12 agents approved, respectively. In our study, the route of administration for registered medications is mainly oral (54 drugs), less common intravenous (6 drugs) and only one (1) is by standard ureteral catheters. Novel drugs have indications for use in monotherapy, combination therapy and adjuvant treatment. Among them, the most common is monotherapy and less common are combination or adjuvant therapies. The monotherapy approach is often used as first-line treatment of non-small cell lung cancer, prostate cancer and thyroid cancer. The combination of approved small molecule drugs with other therapeutic agents is mostly used to treat breast cancer and metastatic melanoma, while surgery and radiotherapy are mostly used in most solid cancers, especially harboring an NRTK gene fusion.

The analysis of 62 active ingredients of new drugs in terms of their chemical structure indicates that almost all (60 active components) of the presented 62 registered drugs are organic compounds, all of which, apart from radiopharmaceutical—**iobenguane I 131** (AZEDRA), contain a heterocyclic system in their structure (59 registered drugs). The few exceptions include lutetium (isotope 177) complex compound, i.e., **lutetium Lu 177 dotatate** (Table 8, item 2), which is formally classified as an inorganic one despite it containing as ligand-a heterocyclic bifunctional chelator derived from DOTA (i.e., 1,4,7,10-Tetraazacyclododecane-1,4,7,10-tetraacetic acid) and one classic inorganic compound, i.e., radium-223 dichloride (Table 8, item 3). Most of all active compounds bear a nitrogen heterocycle ring (60 active ingredients of 60 registered drugs, including the ligand of **lutetium Lu 177 dotatate**). A much smaller number with an oxygen (such as isoxazole, oxazoline, tetrahydrofuran, tetrahydropyran, morpholine, 1,3-benzodioxole and 6-oxa-2-thia-9-azabicyclo[6.2.2]dodecane heterocyclic systems) and sulfur atoms (such as thiazole, thieno[2,3-*d*]pyrimidin and 6-oxa-2-thia-9-azabicyclo[6.2.2]dodecane) heterocycle ring (12 and 6 drugs, respectively). The dominant heterocyclic system is nitrogen monocyclic, non-fused with other rings, pyridine that is present in 17 drugs. In order, 14 drugs contain non-condensed monocyclic pyrimidine ring, 9—piperazine one, 7—pyrazole one, 7—piperidine one, 1—imidazolidine one and 1—azetidine one. A total of 30 drugs have two fused ring heterocyclic systems (bicyclic system), such as quinoline (5 drugs), benzimidazole (4 drugs), quinazoline (3 drugs), indole (2 drugs), indazole (2 drugs), pyrrolo[2,3-*b*]pyridine (2 drugs), pyrido[2,3-*d*]pyrimidin (2 drugs), isoquinoline (1 drug), quinoxaline (1 drug), phthalazine (1 drug), 1,6-naphthyridine (1 drug), pyrrolo[2,3-*d*]pyrimidine (1 drug), pyrrolo[2,1-*f*][1,2,4]triazin (1 drug), pyrazolo[1,5-*a*]pyridine (1 drug), pyrazolo[1,5-*a*]pyrimidin (1 drug), pyrido[4,3-*d*]pyrimidin (1 drug) and thieno[2,3-*d*]pyrimidin (1 drug). Three fused ring heterocyclic systems, such as 5,7,11,13-tetrazatricyclo[7.4.0.02,6]trideca-1,3,6,8-tetraene, 2,3,10-triazatricyclo[7.3.1.05,13]trideca-1,5(13),6,8-tetraen and 3,10-diazatricyclo[6.4.1.04,13]trideca-1,4(13),5,7-tetraen, are present only in three drugs (**pemigatinib**, **talazoparib** and **rucaparib, respectively**). Four or more fused ring heterocyclic systems comprise the main skeleton of six drugs (**lorlatinib** is derivative of 17-oxa-4,5,8,20-tetrazatetracyclo[16.3.1.02,6.010,15]docosa-1(21),2,5,10,12,14,18(22),19-octaene, Alectinib is derivative of 6,11-dihydro-5H-benzo[b]carbazole, **irinotecan** is derivative of 17-oxa-3,13-diazapentacyclo[11.8.0.02,11.04,9.015,20]henicosa-1(21),2,4(9),5,7,10,15(20)-heptaene, **mitomycin** is derivative of 2,5-diazatetracyclo[7.4.0.02,7.04,6]trideca-1(9),11-diene), two of which contain eight fused ring extensive system (i.e., **lurbinectedin** and **trabectedin**). Besides, heterocyclic spiro systems occur in three drugs (i.e, in **apalutamide**, **lurbinectedin** and **trabectedin**). For example, in **apalutamide**, the imidazolidine ring is spiro connected with a tensioned cyclobutane ring. It is worth emphasizing that strongly tensioned cyclopropyl ring is also present in the structure of four (4) drugs (i.e., the multikinase inhibitors, such as **lenvatinib**, **trametinib**, **cabozantinib** and poly (ADP-ribose) polymerases inhibitor-**olaparib**). 

The analysis of heteroatoms of active ingredients of the considered drugs points out that nitrogen atoms are dominant ones (61 compounds), then oxygen—58, halogens (including fluorine, chlorine, bromine and iodine)—43 and sulfur—14 and phosphorus—1. Structurally, the sulfur-containing pharmaceuticals are classified into thiazoles (three), thioketones (two drugs), sulfones (two drugs), sulfonamides (three drugs), thioethers (three drugs) and disulfide (one drug). In the case of halogens, there are one or more atoms in an aliphatic or aromatic system, mainly fluorine and chlorine, less often bromine, and iodine in only two agents, i.e., **trametinib** and **iobenguane I 131**. The introduction of a sulfur-derived functional group or halogen atom is strategic from a medical point of view, as it improves drug selectivity and bioavailability [283,284]. It is interesting to note that one anticancer agent, i.e., **brigatinib**, is a phosphorus-containing compound. A unique structural feature of **brigatinib** is the dimethylphosphine oxide (DMPO) moiety. This functional group has a significant impact on compound potency and selectivity for targeted kinases, especially ALK. Moreover, the highly ionic P=O bond acts as a hydrogen bond acceptor, which increases a molecule’s hydrophilicity, metabolic stability and aqueous solubility [285]. The amide- and urea-containing compounds comprise a large percentage of all new drug approvals. Since amide and urea functional groups are often responsible for the occurrence of biological activity of many natural and synthetic bioactive compounds, they are incorporated into the structures of 27 and 7 new pharmaceuticals, respectively. The donor−acceptor hydrogen bonding capability of amide and urea derivatives plays a key role in their molecular recognition and biological activity [286,287]. The authors also have noticed that nitrile (CN) group can be found in eight new agents for solid tumors, although in general, the nitrile-containing compounds are highly toxic [288]. CN group has the ability to create electron-deficient sites in the structures of these compounds and nonspecific dipole interaction with biological nucleophiles, such as amino acids, nucleic acids and enzymes, that are located in the molecular target [289]. In **alectinib**, for example, CN moiety mainly participates in the formation of hydrogen bonds and the CH/π interactions with various amino acid residues in the active site of ALK. An electron-poor aromatic ring substituted with CN group is, in turn, the pharmacophore of antiandrogenic activity of **darolutamide**, **apalutamide** and **enzalutamide [290]**. Furthermore, 19 small molecules of all approved ones contain chiral active compounds and 1, i.e., **sotorasib**, can exist in both of the two atropoisomeric forms. More specifically, (*R*)-atropoisomer of **sotorasib** is a potent inhibitor of KRAS harboring the G12C mutation [166]. Of the metal coordination complexes of organic ligands, only the radioactive lanthanide Lutetium-177 complex (**lutetium Lu 177 dotatate** as LUTATHERA) entered the antitumor radiotherapy in a considered period. It is worth noting that at that time, no new derivative of platinum complexes was introduced as a drug.

Based on the mechanisms of action, 62 anticancer active compounds are divided into four groups. These include targeted, cytotoxic, radioactive and hormone therapy agents. Most compounds (49) show a targeted mode of action via inhibition of certain molecular targets, which actively support cancer cell growth and spread. Other selective agents belong to the following two classes, hormone therapies (5) and radiopharmaceuticals (3). The hormone therapy drugs block or alter specific hormones that are used by solid tumors, e.g., prostate cancer, for growth. The radiopharmaceuticals, in a selective way, deliver cytotoxic radiation to cancer cells. Only five small molecules are standard chemotherapeutic agents that are characterized by less specific action of disruption of the mitotic and/or DNA replication pathways. Among them, three anticancer active compounds (i.e., **irinotecan**, **mitomycin** and, **trifluridine**) have been used for more than 11 years, but they are now registered in novel formulation under brand names ONIVYDE, JELMYTO and LONSURF, respectively. The new form of these drugs improves their pharmacological parameters, such as stability, solubility and therapeutic index, including increased chemotherapeutic efficacy and reduced adverse effects. Additionally, classifying 62 new pharmaceuticals based on their biological target, there are identified 30 targets necessary for cancer growth and progression. It is also worth noting that first-in-class agents comprise almost one-fourth (15 compounds) of all novel drug approvals. From 2018, two drugs referred to as first in class were approved each year. Moreover, one medication, i.e., LONSURF, is a fixed-dose combination (FDC) drug, which consists of two anticancer active ingredients. LONSURF contains **trifluridine** (inhibitor of thymidylate synthase) and **tripacil** (inhibitor of thymidine phosphorylase) in a molar ratio of 1 to 0.5.

Currently, one of the challenges in developing small molecule agents against solid tumors is drug resistance. This process can be developed by various mechanisms, including drug inactivation, drug target alteration, drug efflux, DNA damage repair and epithelial-mesenchymal transition (EMT) [291]. The changes in drug targets, for example, are the cause of required resistance in the case of signal kinases targeted with small molecule inhibitors (SMIs), such as the first generation of EGFR inhibitors. Hence, there is a strong need to design the next generation of SMIs to overcome drug resistance, as well as to improve their pharmacological properties and safety profile. The use of combination therapy also allows overcoming and reversing drug resistance, reducing adverse effects and improving therapeutic results of treatment. Although a few inhibitor combinations are approved by the FDA, e.g., **vemurafenib** plus **cobimetinib**, **dabrafenib** plus **trametinib** or **encorafenib** plus **binimetinib**, there is still an unsatisfactory number of drugs used for combination therapy. Moreover, the next challenge is to increase the ability of a drug to penetrate the blood–brain barrier (BBB) and thereby access the primary and metastatic tumors of the central nervous system (CNS). **Abemaciclib**, for example, is promising to achieving regression of brain metastases secondary to HR-positive breast cancer. The construction of more comprehensive libraries of kinase genes and their mutations is important to find new key kinase targets or genetic alterations involved in solid cancers growth. This should lead to the development of new strategies, which integrate different disciplines (e.g., chemistry, toxicology, genetics, proteomics, pharmacology) for the design of more selective anticancer agents.

## Figures and Tables

**Figure 1 molecules-27-02259-f001:**
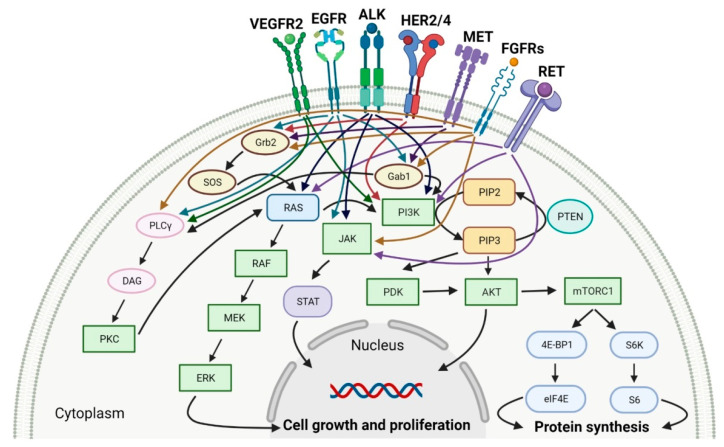
Molecular signal transduction pathways for specific receptor tyrosine kinases (RTKs). **VEGFR2**: vascular endothelial growth factor receptor 2. **EGFR**: epidermal growth factor receptor. **ALK**: anaplastic lymphoma kinase. **HER2/4**: human epidermal growth factor receptor 2 and 4. **MET**: mesenchymal-epithelial transition factor. **FGFRs**: fibroblast growth factor receptors. **RET**: tyrosine kinase rearranged during transfection receptor. **Gab1**: Grb2-associated-binding protein 1. **Grb2**: growth factor receptor-bound protein 2. **SOS**: Son of sevenless. **PLCγ**:. phospholipase C gamma. **DAG**: diacylglycerol. **PKC**: protein kinase C. **RAS**: rat sarcoma viral oncogene homolog. **RAF**: proto-oncogene serine/threonine-protein kinase. **MEK**: mitogen-activated protein kinase kinase. **ERK**: mitogen-activated protein kinase. **PI3K**: phosphatidylinositol 3-kinase. **PIP2**: phosphatidylinositol 4,5-bisphosphate. **PIP3**: phosphatidylinositol-3,4,5-trisphosphate. **PTEN**: phosphatase and tensin homolog deleted on chromosome ten. **PDK**: 3-phosphoinositide-dependent protein kinase. **AKT**: protein kinase B. **mTORC1**: mammalian target of rapamycin complex 1. **4E-BP1**: 4E-binding protein 1. **eIF4E**: eukaryotic translation initiation factor 4E. **S6K**: p70S6 kinase. **S6**: S6 protein. **JAK**: Janus kinase. **STAT**: signal transducer and activator of transcription. Created with BioRender.com based on information in [9,10,11,12,13,14,15].

**Figure 2 molecules-27-02259-f002:**
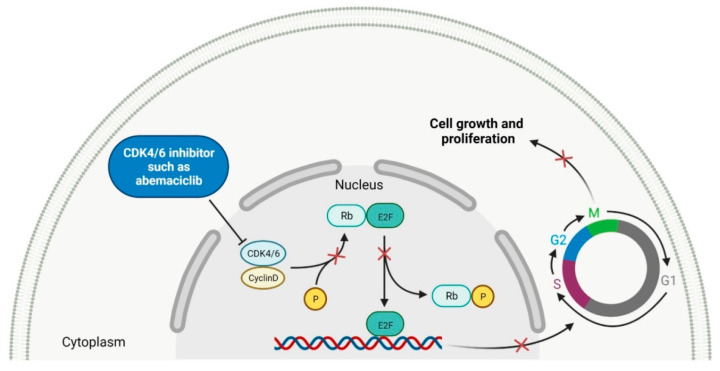
Mechanism of action of CDK4/6 inhibitors (the “x” on the arrows indicates process inhibition). **CDK4/6**: cyclin-dependent kinase 4/6. **P**: phosphate group. **Rb**: retinoblastoma protein. **E2F**: E2 factor. **G1**: first growth phase. **S**: synthesis phase. **G2**: second growth phase. **M**: mitotic phase. Created with BioRender.com based on information in Ref. [60].

**Figure 3 molecules-27-02259-f003:**
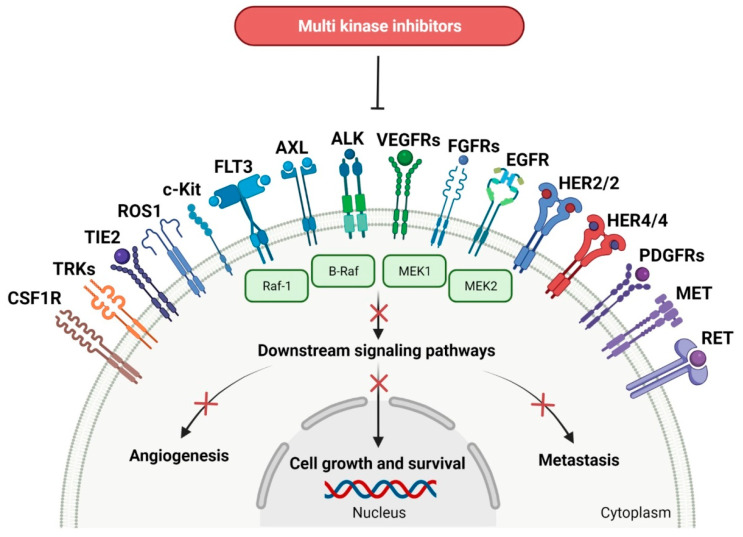
Schematic representation of mode of action of multi-kinase inhibitors that target a set of various related kinases (the “x” on the arrows indicates process inhibition). **CSF1R**: colony-stimulating factor 1 receptor. **TRKs**: tropomyosin receptor tyrosine kinases. **TIE2**: tunica interna endothelial cell kinase 2. **ROS1**: proto-oncogene tyrosine-protein kinase ROS. **c-Kit**: mast/stem cell growth factor receptor. **FLT3**: FMS-like tyrosine kinase-3. **AXL**: AXL receptor tyrosine kinase. **ALK**: anaplastic lymphoma kinase. **VEGFRs**: vascular endothelial growth factor receptors. **FGFRs**: fibroblast growth factor receptors. **EGFR**: epidermal growth factor receptor. **HER2/2**: human epidermal growth factor receptor 2 and 2. **HER4/4**: human epidermal growth factor receptor 4 and 4. **PDGFRs**: platelet-derived growth factor receptors. **RET**: receptor tyrosine kinase rearranged during transfection. **B-Raf**: serine/threonine-protein kinase B-Raf. **Raf-1**: RAF serine/threonine-protein kinase. **MEK1**: mitogen-activated protein kinase kinase 1. **MEK2**: mitogen-activated protein kinase kinase 2. **MET**: mesenchymal-epithelial transition factor. Created with BioRender.com.

**Figure 4 molecules-27-02259-f004:**
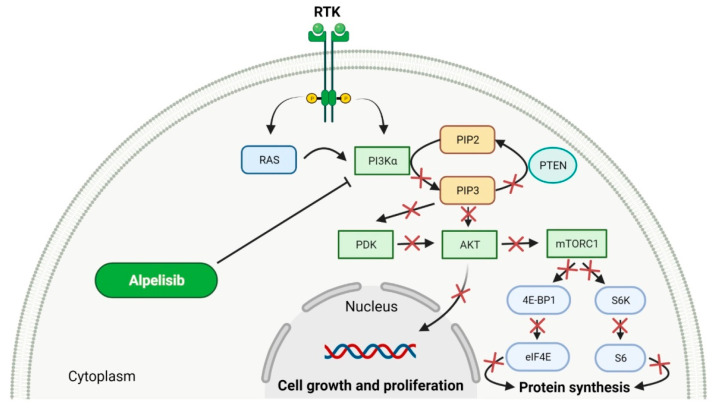
**Alpelisib** inhibition of PI3K/AKT signaling pathway (the “x” on the arrows indicates process inhibition). **RTK**: receptor tyrosine kinase. **RAS**: rat sarcoma viral oncogene homolog. **PI3K-α**: phosphatidylinositol 3-kinase alpha. **PIP2**: phosphatidylinositol 4,5-bisphosphate. **PIP3**: phosphatidylinositol-3,4,5-trisphosphate. **PTEN**: phosphatase and tensin homolog deleted on chromosome ten. **PDK**: 3-phosphoinositide-dependent protein kinase. **AKT**: protein kinase B. **mTORC1**: mammalian target of rapamycin complex 1. **4E-BP1**: 4E-binding protein 1. **eIF4E**: eukaryotic translation initiation factor 4E. **S6K**: p70S6 kinase. **S6**: S6 protein. Created with BioRender.com based on information in Ref. [159].

**Figure 5 molecules-27-02259-f005:**
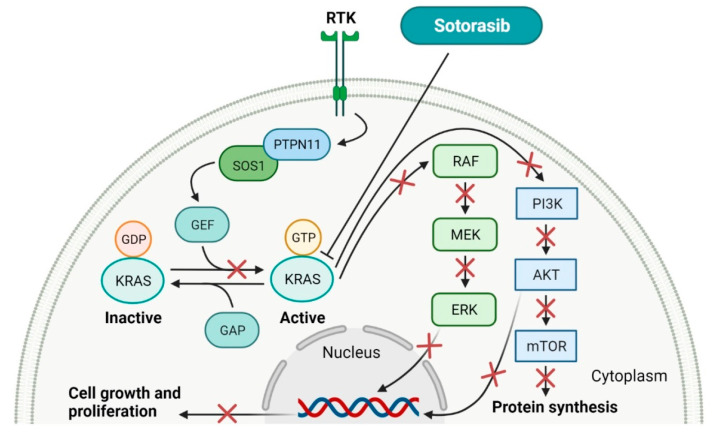
Overview of mechanism of action of **sotorasib** (the “x” on the arrows indicates process inhibition). **RTK**: tyrosine kinase receptor. **PTPN11**: tyrosine-protein phosphatase non-receptor type 11. **SOS1**: Son of sevenless. **GEF**: guanine nucleotide exchange factor. **GAP**: GTPase-activating protein. **KRAS**: GTPase KRas. **GDP**: guanosine-5’-diphosphate. **GTP**: guanosine-5’-triphosphate. **RAF**: proto-oncogene serine/threonine-protein kinase. **MEK**: mitogen-activated protein kinase kinase. **ERK**: mitogen-activated protein kinase. **PI3K**: phosphatidylinositol 3-kinase. **AKT**: protein kinase B. **mTOR**: mammalian target of rapamycin. Created with BioRender.com based on information in Ref. [167].

**Figure 6 molecules-27-02259-f006:**
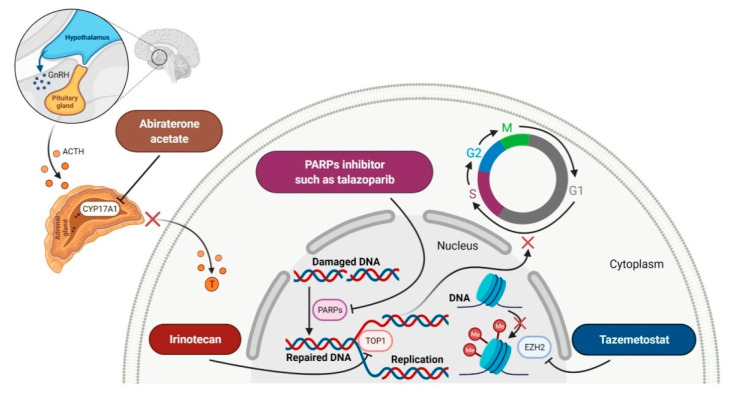
The four types of enzyme inhibitors and their mode of targeted solid cancers treatment (the “x” on the arrows indicates process inhibition). **GnHR**: gonadotropin-releasing hormone. **ACTH**: adrenocorticotropic hormone. **CYP17A1**: steroid 17alpha-monooxygenase. **T**: testosterone. **DNA**: deoxyribonucleic acid. **PARPs**: poly (ADP-ribose) polymerases. **TOP1**: topoisomerase 1. **EZH2**: enhancer of zeste homolog 2. **Me**: methyl group. **G1**: first growth phase. **S**: synthesis phase. **G2**: second growth phase. **M**: mitotic phase. Created with BioRender.com.

**Figure 7 molecules-27-02259-f007:**
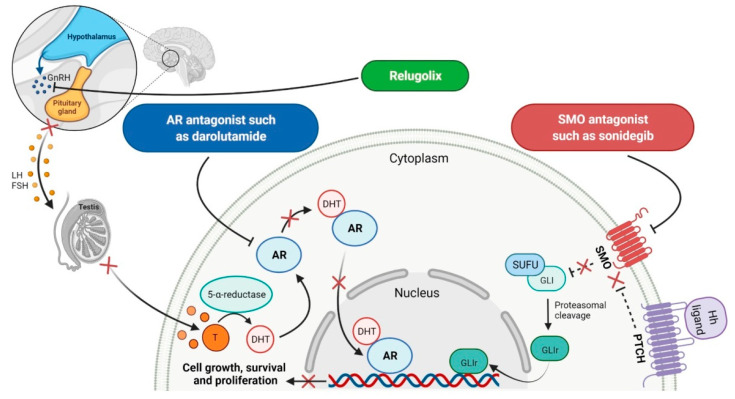
Mechanism of action of receptor antagonists in the AR, GnRH-r and SMO receptor signaling pathways (the “x” on the arrows indicates process inhibition). **GnRH**: gonadotropin-releasing hormone. **LH**: luteinizing hormone. **FSH**: follicle stimulating hormone. **T**: testosterone. **DHT**: dihydrotestosterone. **AR**: androgen receptor. **HH**: hedgehog. **PTCH**: patched receptor. **SMO**: smoothened receptor. **SUFU**: suppressor of fused protein. **GLI**: glioma-associated oncogene protein. **GLIr**: repressor form of GLI. Created with BioRender.com based on information in Refs. [203,204].

**Table 2 molecules-27-02259-t002:** Features of the cyclin-dependent kinase inhibitors approved as drugs by the Food and Drug Administration (FDA) from 2011 to 2022. The order of drugs is tabulated in order of most recent to oldest registration date. A generic name of a drug is an international nonproprietary name (INN).

No.	Generic Name of Drug	Brand Name and Company	First FDA/EMA Approval Date	Structure	Molecular Target	Route of Administration	Indication	Adverse Effects	Ref.
1	Abemaciclib	VERZENIO Eli Lilly and Company, Indianapolis, IN, USA	FDA: 28 September 2017 EMA: 27 September 2018	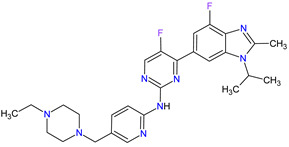	CDK4 ^1^, CDK6 ^2^	Oral	Breast Cancer	Diarrhea, fatigue, nausea, decreased appetite, abdominal pain, neutropenia, vomiting, infections, anemia, headache, thrombocytopenia, leucopenia	[67,68]
2	Ribociclib	KISQALI Novartis Pharmaceuticals Corporation, Basel, Switzerland	FDA: 13 March 2017 EMA: 22 August 2017	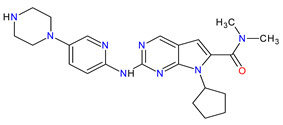	CDK4 ^1^, CDK6 ^2^	Oral	Breast Cancer	Neutropenia, nausea, infections, fatigue, diarrhea	[69,70]
3	Palbociclib	IBRANCE Pfizer Inc., New York City, NY, USA	FDA: 3 February 2015 EMA: 9 November 2016	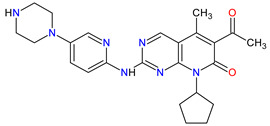	CDK4 ^1^, CDK6 ^2^	Oral	Breast Cancer	Neutropenia, leukopenia, fatigue, anemia, nausea, arthralgia, alopecia, diarrhea, hot flush	[71,72]

^1^ **CDK4**: cyclin-dependent kinase 4. ^2^ **CDK6**: cyclin-dependent kinase 6.

**Table 4 molecules-27-02259-t004:** Features of the phosphatidylinositol-3 kinase-α inhibitor approved as drugs by the Food and Drug Administration (FDA) from 2011 to 2022. A generic name of a drug is an international nonproprietary name (INN).

No.	Generic Name of Drug	Brand Name and Company	First FDA/EMA Approval Date	Structure	Molecular Target	Route of Administration	Indication	Adverse Effects	Ref.
1	Alpelisib	PIQRAY Novartis Pharmaceuticals Corporation, Basel, Switzerland	FDA: 24 May 2019 EMA: 27 July 2020	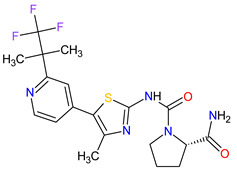	PI3K-α ^1^	Oral	Breast Cancer	Hyperglycemia, diarrhea, rash, nausea, fatigue, decreased appetite, stomatitis	[160,161]

^1^ **PI3K-α**: phosphatidylinositol 3-kinase alpha.

**Table 5 molecules-27-02259-t005:** Features of the GTPase KRas inhibitor approved as drugs by the Food and Drug Administration (FDA) from 2011 to 2022. A generic name of a drug is an international nonproprietary name (INN).

No.	Generic Name of Drug	Brand Name and Company	First FDA/EMA Approval Date	Structure	Molecular Target	Route of Administration	Indication	Adverse Effects	Ref.
1	Sotorasib	LUMAKRAS Amgen Inc., Thousand Oaks, CA, USA	FDA: 28 May 2021 EMA: Not approved	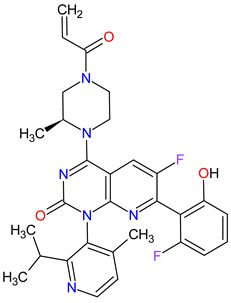	KRAS ^1^	Oral	Non-Small Cell Lung Cancer	Decreased lymphocytes, and hemoglobin, diarrhea, increased aspartate aminotransferase, alanine aminotransferase, and alkaline phosphatase, musculoskeletal pain, decreased calcium, nausea, fatigue, hepatotoxicity, cough	[168]

^1^ **KRAS**: GTPase KRas.

**Table 10 molecules-27-02259-t010:** Features of the fixed-dose combination drug approved as drugs by the Food and Drug Administration (FDA) from 2011 to 2021. A generic name of a drug is an international nonproprietary name (INN).

No.	Generic Name of Drug	Brand Name and Company	First FDA/EMA Approval Date	Structure	Molecular Target	Route of Administration	Indication	Adverse Effects	Ref.
1	Trifluridine + Tipiracil	LONSURF Taiho Oncology, Inc., Princeton, NJ, USA	FDA: 22 September 2015 EMA: 26 April 2016	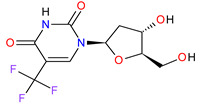	TS ^1^	Oral	Colorectal Cancer, Gastric Cancer	Anemia, neutropenia, fatigue/asthenia, nausea, thrombocytopenia, decreased appetite, diarrhea, vomiting, pyrexia	[268,269,270]
				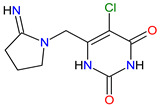	TP ^2^				

^1^ **TS**: thymidylate synthase. ^2^ **TP**: thymidine phosphorylase.

**Table 11 molecules-27-02259-t011:** Features of several examples of potential anticancer drugs in the pipeline.

No.	Name/Symbol	Company	Phase of Development	Structure	Molecular Target	Route of Administration	Indication	Ref.
1	Adagrasib (MRTX849)	Mirati Therapeutics, Inc., San Diego, CA, USA	Phase 3	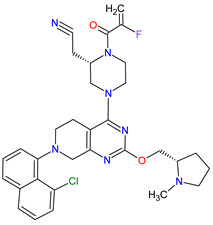	KRAS ^1^	Oral	Non-Small Cell Lung Cancer	[277]
2	Giredestrant (GDC-9545)	Genentech, Inc., South San Francisco, CA, USA	Phase 3	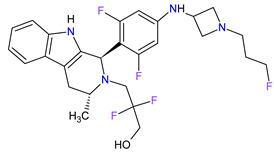	ER ^2^	Oral	Breast Cancer	[274]
3	Varlitinib (ARRY-334543)	ASLAN Pharmaceuticals, Menlo Park, CA, USA	Phase 3	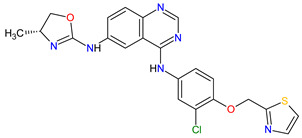	EGFR ^3^, HER2 ^4^	Oral	Gastric Cancer	[276,278]
4	ARV-471	Arvinas, Inc., New Haven, CT, USA Pfizer Inc., New York City, NY, USA	Phase 2	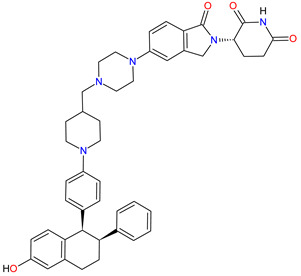	ER ^2^	Oral	Breast Cancer	[273]
5	Pemrametostat (GSK3326595)	GlaxoSmithKline, London, UK	Phase 2	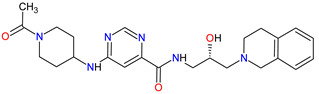	PRMT5 ^5^	Oral	Breast Cancer	[272,279]
6	Adavosertib (AZD1775)	AstraZeneca, Cambridge, UK	Phase 2	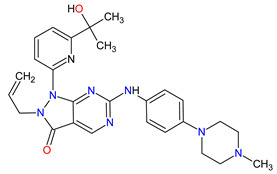	WEE1 ^6^	Oral	Solid Tumors	[271,280]
7	Ipatasertib (GDC-0068)	Genentech, Inc., South San Francisco, CA, USA	Phase 2	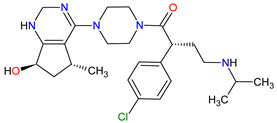	AKT ^7^	Oral	Gastric Cancer	[275]

^1^ **KRAS**: GTPase KRas. ^2^ **ER**: estrogen receptor. ^3^ **EGFR**: epidermal growth factor receptor. ^4^ **HER2**: human epidermal growth factor receptor 2. ^5^ **PRMT5**: protein arginine methyltransferase 5. ^6^ **WEE1**: WEE1 G2 checkpoint kinase. ^7^ **AKT**: protein kinase B.

## Data Availability

Not applicable.

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
