# Peer review of "FDA-Approved Small Molecule Compounds as Drugs for Solid Cancers from Early 2011 to the End of 2021"

_molecules, 2022, doi:10.3390/molecules27072259_

Round 1

Reviewer 1 Report

 In this review manuscript, the small molecule drugs registered by the Food and Drug Administration (FDA) for solid cancers treatment between 2011 and 2022 were comprehensively summarized and discussed.   The contents presented in this review are undoubtedly useful for the medicinal chemists to further design and develop more specific targeted agents for treatment of the solid tumors. This reviewer suggests its acceptance in its current form. 

Two tiny suggestions: 1. The chemical structures are not uniformly formatted. Some are presented with bigger atom symbols while some has been stretched. It is suggested that some modification to these structure is made. 2. The diagram should be downsized as the current large size is not compatible with the text.

Author Response

We would like to thank Reviewer for taking the time and effort necessary to review our manuscript and for recommend it for publication. Below, there are our explanations and response to the Reviewer remarks:

Point 1: The chemical structures are not uniformly formatted. Some are presented with bigger atom symbols while some has been stretched. It is suggested that some modification to these structure is made.

Response 1: We attempted to equalize the size and improve the view of the structures shown in the tables. The tables were slightly reformatted to make the compound structures similar in size, but this was not always successful due to the limited width of the table box.

Point 2: The diagram should be downsized as the current large size is not compatible with the text.

Response 2: It was corrected. The size of the figures has been adjusted to the margins of the page.

Reviewer 2 Report

Manuscript 1644224 review the small molecule compounds for the treatment of cancer approved by the FDA from 2011 to 2022. This review enables the readers to understand the trends in drug development, and the concept of this study is meaningful for the clinical interest.

  1. There are several pipeline small molecule compounds for cancer treatment. The authors should mention it. Furthermore, the authors should show the future direction of small molecule compounds, including the challenges to overcome in developing.

Author Response

We would like to thank the Reviewer for taking the time and effort necessary to review our manuscript and for all the valuable comments and suggestions, which helped us to improve it. Below, there are our explanations and response to the Reviewer remarks:

Point 1: There are several pipeline small molecule compounds for cancer treatment. The authors should mention it.

Response 1: We have included some examples of the potential anticancer drugs in the pipeline in Table 11 (pages 48-49). We chose small molecule compounds showing the most promising antitumor activity in Phase II and III of clinical trials.

Point 2: Furthermore, the authors should show the future direction of small molecule compounds, including the challenges to overcome in developing.

Response 2: We have outlined in Conclusions the challenges and future directions in designing and developing of small molecule agents against solid tumors.